Corrected: Publisher correction

# Pharmacological reactivation of MYC-dependent apoptosis induces susceptibility to anti-PD-1 immunotherapy

Heidi M. Haikala [1,16], Johanna M. Anttila[1], Elsa Marques[1], Tiina Raatikainen[1], Mette Ilander[2], Henna Hakanen[2], Hanna Ala-Hongisto[1], Mariel Savelius[1], Diego Balboa [3], Bjoern Von Eyss [4], Vilja Eskelinen[1], Pauliina Munne[1], Anni I. Nieminen[5], Timo Otonkoski [3], Julia Schüler[6], Teemu D. Laajala[7,8], Tero Aittokallio [7,8], Harri Sihto [9,10], Johanna Mattson[10], Päivi Heikkilä[11], Marjut Leidenius[12], Heikki Joensuu[9,10], Satu Mustjoki [2], Panu Kovanen[13], Martin Eilers [14], Joel D. Leverson[15] & Juha Klefström[1]

Elevated MYC expression sensitizes tumor cells to apoptosis but the therapeutic potential of this mechanism remains unclear. We find, in a model of MYC-driven breast cancer, that pharmacological activation of AMPK strongly synergizes with BCL-2/BCL-X$_L$ inhibitors to activate apoptosis. We demonstrate the translational potential of an AMPK and BCL-2/BCL-X$_L$ co-targeting strategy in ex vivo and in vivo models of MYC-high breast cancer. Metformin combined with navitoclax or venetoclax efficiently inhibited tumor growth, conferred survival benefits and induced tumor infiltration by immune cells. However, withdrawal of the drugs allowed tumor re-growth with presentation of PD-1+/CD8+ T cell infiltrates, suggesting immune escape. A two-step treatment regimen, beginning with neoadjuvant metformin+venetoclax to induce apoptosis and followed by adjuvant metformin+venetoclax+anti-PD-1 treatment to overcome immune escape, led to durable antitumor responses even after drug withdrawal. We demonstrate that pharmacological reactivation of MYC-dependent apoptosis is a powerful antitumor strategy involving both tumor cell depletion and immunosurveillance.

---

[1] Cancer Cell Circuitry Laboratory, Research Programs Unit/Translational Cancer Biology and Medicum, University of Helsinki, P.O. Box 63 Street address: Haartmaninkatu 8, 00014 Helsinki, Finland. [2] Hematology Research Unit Helsinki, Department of Clinical Chemistry and Hematology, University of Helsinki and Helsinki University Hospital Comprehensive Cancer Center, Haartmaninkatu 8, 00290 Helsinki, Finland. [3] Research Programs Unit/Molecular Neurology, Biomedicum Stem Cell Center, University of Helsinki, Haartmaninkatu 8, 00290 Helsinki, Finland. [4] Leibniz Institute of Age Research, Fritz Lipmann Institute e.V, Beutenbergstraße 11, 07745 Jena, Germany. [5] Department of Biosciences and Institute of Biotechnology, University of Helsinki, Viikinkaari 5, 00790 Helsinki, Finland. [6] Oncotest GmbH, (Now part of Charles River Laboratories Inc, 251 Ballardvale St, Wilmington, MA 01887, USA), Freiburg, Germany. [7] Institute for Molecular Medicine Finland (FIMM), University of Helsinki, Tukholmankatu 3, 00290 Helsinki, Finland. [8] Department of Mathematics and Statistics, University of Turku, Vesilinnantie 5, 20500 Turku, Finland. [9] Research Programs Unit / Translational Cancer Biology & Medicum, University of Helsinki, P.O. Box 63 (Street address: Haartmaninkatu 8), 00290 Helsinki, Finland. [10] Department of Oncology, University of Helsinki and Helsinki University Hospital, Haartmaninkatu 4, 00290 Helsinki, Finland. [11] Department of Pathology, University of Helsinki and Helsinki University Hospital, Haartmaninkatu 3, 00290 Helsinki, Finland. [12] Breast Surgery Unit, Helsinki University Hospital, Kasarmikatu 11-13, 00290 Helsinki, Finland. [13] Department of Pathology, HUSLAB and Haartman Institute, University of Helsinki and Helsinki University Hospital, Haartmaninkatu 3, 00290 Helsinki, Finland. [14] Theodor Boveri Institute and Comprehensive Cancer Center Mainfranken, Biocenter, University of Würzburg, Am Hubland D-970074, Germany. [15] Oncology Development, AbbVie, Inc., 1 N Waukegan Road, North Chicago, IL 60064, USA. [16] Present address: Department of Medical Oncology, Dana-Farber Cancer Institute and Harvard Medical School, 360 Longwood Ave, 02215 Boston, MA, USA. These authors contributed equally: Johanna M. Anttila, Elsa Marques, Tiina Raatikainen. Correspondence and requests for materials should be addressed to J.K. (email: Juha.Klefstrom@helsinki.fi)

MYC is a multifunctional oncogenic transcription factor that is frequently overexpressed in cancer. The *MYC* gene locus is amplified in about 16% of all breast tumors and about one-third of breast tumors overexpress *MYC* mRNA[1–3]. In a genetic landscape study of breast cancer, *MYC* stands out as one of the seven key driver cancer genes[4]. MYC protein expression is also elevated via altered post-translational mechanisms and, altogether, about half of breast cancers display elevated MYC protein expression[5]. *MYC* over-expression and amplification are associated with breast tumor progression and increased risk of relapse and death[3,6].

When overexpressed, MYC can promote transcription, not only via its canonical binding sites, but also by occupying low affinity promoters. Such "promoter invasion" may endow cells with new tumor-specific phenotypes[7], including insensitivity to proliferation-restricting signals, altered cell metabolism in support of continuous growth, and effects on the tumor micro-environment[8]. However, deregulated MYC expression also creates cancer vulnerabilities that can be exploited therapeutically. For example, the effects of oncogenic MYC on cell metabolism, host-microenvironment communication, and immunoregulation have all been considered as potential nodes for targeting MYC indirectly[9–12].

Perhaps the most interesting vulnerability from a therapeutic standpoint is the strong pro-apoptotic activity of MYC[13,14], which involves induction or activation of pro-apoptotic BCL-2 family members, such as BIM, BAK, and BAX, or reduction of anti-apoptotic members, like BCL-2 and BCL-$X_L$. Independently or in combination, these changes can "prime" and activate the intrinsic (mitochondrial) pathway of programmed cell death[13]. Findings in mouse tumor models have indicated that MYC's apoptotic function normally presents a major roadblock to tumor formation[15], but that overexpression of BCL-2 or BCL-$X_L$ or loss-of-p53 efficiently rescues tumors from apoptosis without reducing the tumor-promoting functions of MYC[13,16].

The development of small-molecule BH3 mimetics, which bind and neutralize anti-apoptotic BCL-2 family proteins, has motivated attempts to therapeutically reactivate the apoptotic potential of MYC in tumors. Optimally, pharmacological reactivation of MYC-dependent apoptosis would eradicate tumors without harming normal cells expressing physiological levels of MYC. BH3 mimetics such as the BCL-2/BCL-$X_L$ inhibitor ABT-737, its orally bioavailable derivative ABT-263/navitoclax, or BCL-2-specific ABT-199/venetoclax, have shown an ability to restrain lymphomagenesis in Eμ-Myc mouse models of lymphoma. Furthermore, improved activity has been obtained by combining BH3 mimetics with standard chemotherapy[17], proteasome inhibitors, or histone deacetylase inhibitors[18,19]. These findings, while encouraging, underscore the pressing need to find efficient mechanism-based approaches to fully reactivate apoptosis in cancer cells and maximize therapeutic benefit.

We explored the antitumor effects of BCL-2/BCL-$X_L$ inhibition using ABT-737 in a mouse model of Myc-driven breast cancer. Although ABT-737 was sufficient to induce apoptosis and reduce tumor growth as monotherapy, it failed to provide survival benefit. Our efforts to identify optimal companion drugs unexpectedly exposed strong apoptotic synergy with agents that induce AMP-activated protein kinase (AMPK) activation. Robust activation of MYC-associated apoptosis by combined BCL-2/BCL-$X_L$ inhibition and AMPK activation suppressed tumor growth, offered survival benefits, and increased the infiltration and activity of immune cells in the tumor tissue. Tumors that grew post-treatment were found to be infiltrated by PD-1-positive cytotoxic T cells, consistent with the emergence of post-therapy immune exhaustion. More durable therapeutic effects were obtained when BCL-2/BCL-$X_L$ inhibition and AMPK activation in the adjuvant setting were supplemented with anti-PD-1 therapy. These findings demonstrate that MYC-induced apoptotic sensitivity is an actionable tumor vulnerability, especially when combined with immune checkpoint blockade.

## Results

**MYC and the anti-apoptotic BCL-2 proteins in breast cancer.** To determine whether primary breast cancer could be targeted by a therapeutic strategy that reactivates MYC's apoptotic potential via BH3 mimetics, we assessed the expression of MYC, BCL-2, BCL-$X_L$, and MCL-1 using a tissue microarray (TMA) of 231 primary breast cancer samples. Immunohistochemistry revealed a widespread nuclear staining of MYC in almost half of the samples (Fig. 1a) (MYC-high, >50% cells positive). Surprisingly, >40% of the samples were mostly negative for MYC expression (MYC-low; <20% cells positive), and only a minor fraction of the samples fell in-between the MYC-high and MYC-low categories. Thus, MYC expression is noticeably dichotomous in primary breast cancer. The expression levels of cytosolic BCL-2 family proteins were defined as negative, weak, intermediate, or strong, and the results of both the MYC and the BCL-2 family stainings were analyzed across the major breast cancer subtypes. BCL-2, BCL-$X_L$, and MCL-1 proteins were expressed at intermediate/strong levels in ≥37% of cases in each subtype (Supplementary Figure 1A). More than half of the breast cancers with MYC-high status expressed a high level of BCL-2, BCL-$X_L$, or MCL-1 (Fig. 1b). Thus, high expression levels of MYC and anti-apoptotic BCL-2 proteins are common features in breast cancer.

**BH3 mimetic ABT-737 inhibits Myc-driven mammary tumor growth.** To study whether simultaneous neutralization of Bcl-2 and Bcl-$X_L$ by the prototype BH3 mimetic ABT-737 reactivates Myc-associated apoptosis in vivo, we utilized an autochthonous WapMyc mouse model of breast cancer. In this model, lactogenic hormones activate the Wap promoter during late pregnancy, leading to high Myc expression in the luminal cells of the mammary gland and the formation of solitary adenocarcinomas within 2–3 months[20,21]. WapMyc tumor cells express high levels of Bcl-2, Bcl-$X_L$ and Mcl-1 (Fig. 1d), and are sensitive to ABT-737 ex vivo (Fig. 1e, Supplementary Figure 1B–C). Tumor-bearing WapMyc mice received daily administration of ABT-737 for 21 days and were followed up for another 21 days (Fig. 1c). ABT-737 treatment increased apoptosis in tumors, although overall apoptosis was low (<5%), perhaps due to the rapid clearance of apoptotic cells in live tissues (Fig. 1f, Supplementary Figure 1D). ABT-737 inhibited tumor growth during the first 12 days of treatment (Fig. 1g), after which time the first animals had to be killed. Therefore, at later time points the average tumor volume does not indicate tumor growth rates reliably (Fig. 1g left, gray area). To estimate the tumor growth rates throughout the experimental period, we used a mixed-effects modeling framework that excludes killed animals from the experiment. While ABT-737 inhibited tumor growth over the whole 21-day treatment period (Supplementary Figure 1E), there was no statistically significant difference in the overall survival between the control and ABT-737-treated groups (Fig. 1g, right). Interestingly, lung metastases were observed in only 27% of the ABT-737-treated mice, compared to 50% of mice in the control group (Fig. 1h).

To test the efficacy of ABT-737 in parallel cohorts, we developed an orthotopic WapMyc tumor syngraft model. Tumor cell suspensions from donor FVB mice were grafted bilaterally to cleared fat pads of the syngeneic host mice. The syngrafted tumors appeared 2 weeks after the transplantations and retained their original histological adenocarcinoma phenotype with high Myc expression (Fig. 1i–j). ABT-737 treatment-induced apoptosis

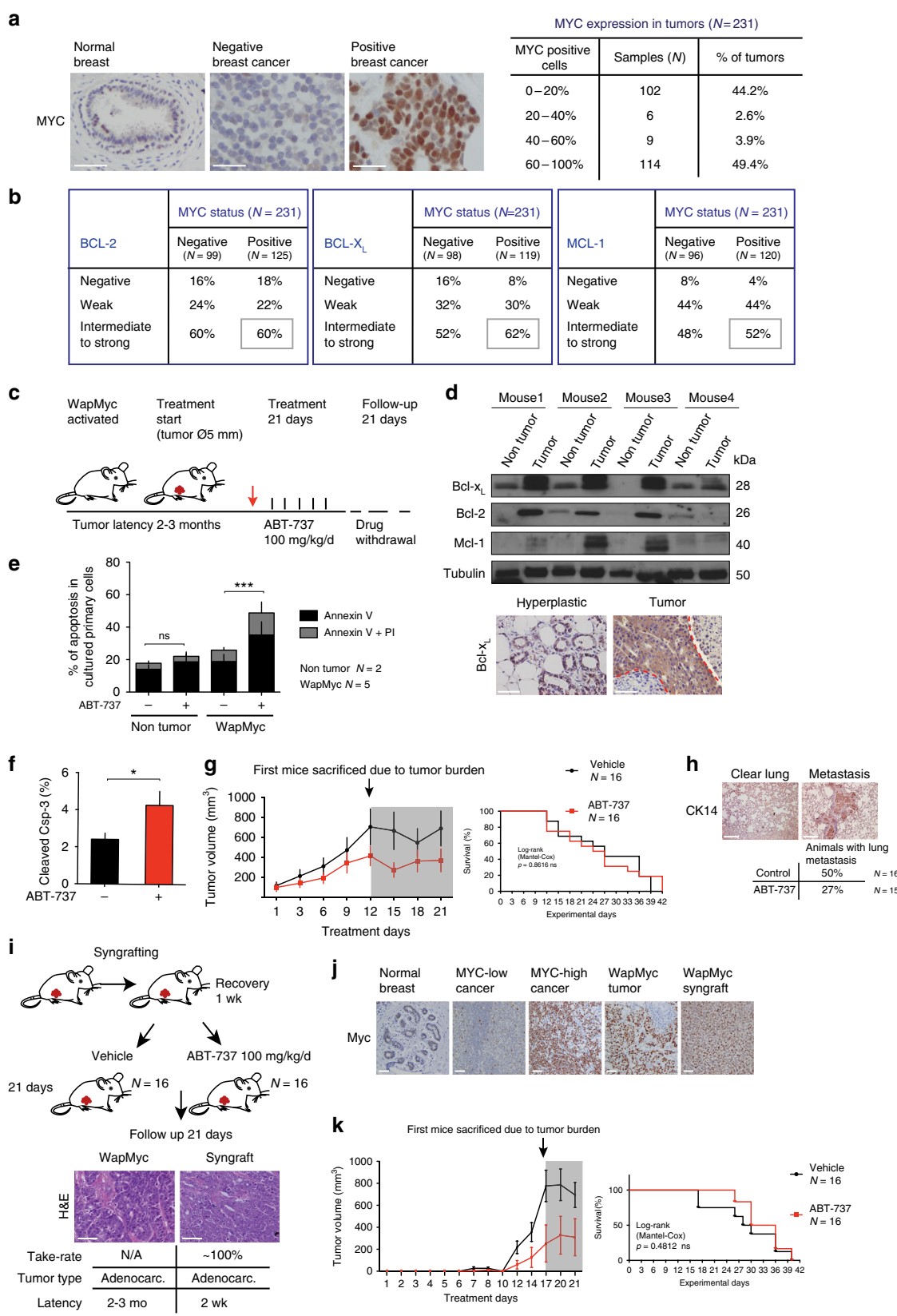

in these tumors (Supplementary Figure 1F–G) and slowed tumor growth at early stages of tumor formation (Fig. 1k, left). Once again there was no difference in the overall survival between the control and the ABT-737-treated groups (Fig. 1k, right).

In summary, ABT-737 induces apoptosis, inhibits the growth of primary tumor and its dissemination to distant tissues. However, these effects did not provide clear survival benefits over time.

**Fig. 1** Inhibition of anti-apoptotic BCL-2 proteins by ABT-737 curbs WapMyc-driven mammary tumor growth. **a** Nuclear expression of MYC in breast cancer. Representative images from a breast cancer tissue microarray (TMA) immunostained for MYC. Samples that were missing from the TMA slide or had insufficient clinical diagnosis data, or had inadequate technical quality were excluded from all analyses. **b** Cross-tabulation of MYC expression against BCL-2, BCL-$X_L$, or MCL-1 expression status. Gray boxes: Double-positive samples. **c** ABT-737 treatment protocol for WapMyc-induced autochthonous mammary adenocarcinoma. Red arrow: start of the treatment. **d** Upper: Western blot analysis of anti-apoptotic BCL-2 proteins in parallel non-tumor and tumor glands dissected from the same WapMyc mouse ($N = 4$ mice). Tubulin: Loading control. Lower: Immunostaining of BCL-$X_L$ in a hyperplastic gland and a tumor gland. Tumor–stroma border indicated with red dotted line. **e** Flow cytometric quantification of apoptosis in primary cultures. Epithelial cells were isolated from tumor or non-tumor glands, treated 24 h with 1 μM ABT-737 and stained with Annexin V/PI. Triplicate experiments performed on control ($N = 2$) or WapMyc tumor cell isolates ($N = 5$). Student's $t$-test (unpaired), SD. **f** Quantification of apoptosis in tumor tissue. Positive cells scored from 16 vehicle and 16 ABT-737-treated tumors. Student's $t$-test (unpaired), SD. **g** Left: Tumor growth during the 21-day treatment period. Arrow: First mice killed due to tumor burden (∅ > 2 cm). Right: Survival of the mice. **h** ABT-737 treatment decreases incidence of lung metastases in WapMyc tumor bearing mice, Student's $t$-test (unpaired). **i** ABT-737 treatment protocol for orthotopically syngrafted WapMyc tumors. **j** MYC expression pattern in normal human breast tissue, human breast tumors, and endogenous & syngrafted WapMyc mouse mammary tumors. **k** Left: Tumor growth during the 21-day treatment period. Arrow: The first mice sacrificed due to tumor burden (∅ > 2 cm). Right: Survival of the mice

**Co-targeting AMPK and BCL-2/-$X_L$ sensitizes to MYC apoptosis.** To identify drugs that could enhance the apoptotic effect of BCL-2/BCL-$X_L$ inhibition, while still preserving MYC-dependence, we analyzed the effect of ABT-737 in combinations with small-molecules that target cancer survival pathways (Fig. 2a). For the assays, we employed non-transformed MCF10A mammary epithelial cells expressing a conditionally active form of MYC (MycER)[14]. The apoptotic potential of single-agent and combination treatments was tested with and without MYC activation (Fig. 2b–c). All tested compounds potentiated the apoptotic action of ABT-737 but the level of MYC-dependence varied. Combinations with BEZ-235 and MK2206 were active regardless of MYC status (Fig. 2c) and not investigated further. Interestingly, a strict MYC-dependency was observed for apoptosis induced by Nutlin-3a and PRIMA-1, which both act by inducing p53 activity, and A-769662, a compound that activates AMPK by an allosteric mechanism (Fig. 2d). These observations are consistent with earlier findings showing that the MYC-dependent pro-apoptotic action of AMPK is coupled with p53 in MCF10A cells[22]. A-769662 did not induce cell death without sensitization by MYC and ABT-737, even at high concentrations (Fig. 2e–f). Metformin, a commonly used type II diabetes drug that also activates AMPK, induced similar MYC-dependent apoptosis when combined with ABT-737 (Fig. 2g–h).

To further examine whether high endogenous expression of MYC induces apoptotic sensitivity, we employed a dCas9VP192-assisted transcriptional activation system to upregulate MYC expression from its own locus[23]. In this system, guide-RNAs targeted to MYC promoter-proximal areas recruit a fusion protein comprising dead CRISPR-associated protein (dCas9) and repeats of Herpes simplex virus protein-16 transactivation domain (VP192) (Fig. 2i). Chemical activation of the MYC promoter-guided dCas9VP192-induced endogenous MYC expression (Fig. 2i, Supplementary Figure 2A) and sensitized MCF10A cells to apoptosis by ABT-737+A-769662 (Fig. 2j). Together, these results establish a role for MYC in rendering cells sensitive to combined AMPK activation and BCL-2/BCL-$X_L$ inhibition.

**AMPK-induced BIM mediates apoptosis.** We asked whether AMPK sensitizes to ABT-737 by reducing the expression of anti-apoptotic BCL-2 family proteins like MCL-1, a known resistance factor for BH3 mimetics[24]. However, A-769662 did not alter the levels of BCL-2, BCL-$X_L$, or MCL-1 (Fig. 3a, Supplementary Figure 2B). To explore the role of pro-apoptotic BCL-2 proteins, we silenced the expression of BH3-only proteins in MCF10A cells prior to inducing apoptosis with the well-established synthetic-lethal combination of MYC and TRAIL (Fig. 3b, Supplementary Figure 2C)[14,25]. Apoptosis was completely abolished by the loss of

BID, consistent with its role as the main mediator of TRAIL-induced apoptosis. The only other BH3-only protein as crucial for MYC-TRAIL-induced apoptosis was BIM (Fig. 3b), which has been suggested to drive MYC-dependent apoptosis also in other studies[26] (Supplementary Figure 2D).

We next studied whether AMPK activation upregulates BIM. MCF10A cells express two BIM isoforms, corresponding to alternatively spliced isoforms, $BIM_L$ and $BIM_{EL}$. These isoforms differ in their properties, $BIM_L$ being a more potent activator of apoptosis than $BIM_{EL}$[27]. MYC induced a slight upregulation of both isoforms, consistent with earlier findings[26]. Notably, A-769662 induced a strong upregulation of $BIM_L$ independent of MYC expression or the addition of ABT-737 (Fig. 3c). Furthermore, silencing of BIM specifically abolished the AMPK-dependent enhancement of apoptosis, which establishes BIM as a major contributor to this mechanism (Fig. 3d, compare to Fig. 2f).

To examine the relevance of these findings in cancer, we investigated the status of AMPK activity and BIM in a set of clinical breast cancer samples with known MYC status (Fig. 1a). Although these analyses suggested no correlation between MYC and BIM levels or MYC and AMPK activity (detected by phosphorylated Acetyl-CoA Carboxylase, pACC) (Fig. 3e), they revealed a strong correlation between AMPK activity and BIM levels. These results are consistent with the data in Fig. 3c and suggest that chronically high MYC levels may not be compatible with high AMPK activity and high BIM levels in tumors, as the combination would likely induce apoptosis (Fig. 3f). This idea led to the hypothesis that activation of AMPK could sensitize MYC-driven tumors to BH3 mimetics.

**Apoptosis induction in MYC-positive breast cancer explants.** We next examined whether the AMPK and BCL-2/BCL-$X_L$ co-targeting strategy triggers MYC-dependent apoptosis in human breast cancer tissue. For this purpose, we established a three-dimensional (3D) Patient-Derived Explant culture (PDEc) model of primary breast cancer (Supplementary Figure 3A–C). Ten breast tumors, three tumor-adjacent areas (adjacent), and three non-tumor control tissues (reduction mammoplasty) were processed and analyzed as in Fig. 4a. Consistent with our earlier findings suggesting dichotomous expression of MYC in primary breast cancer samples (Fig. 1a), five out of ten tumor samples were defined as MYC-high; whereas, the other half of the samples were MYC-low (Fig. 4b). Comparison of the MYC status in two original tumor and adjacent tissue samples and their explant derivatives demonstrated that PDEc culture conditions preserve their original tumor sample-specific nuclear MYC status (Fig. 4c–d).

We tested the pro-apoptotic potential of ABT-737+A-769662 combination in PDEcs (Fig. 4e). No apoptotic responses were

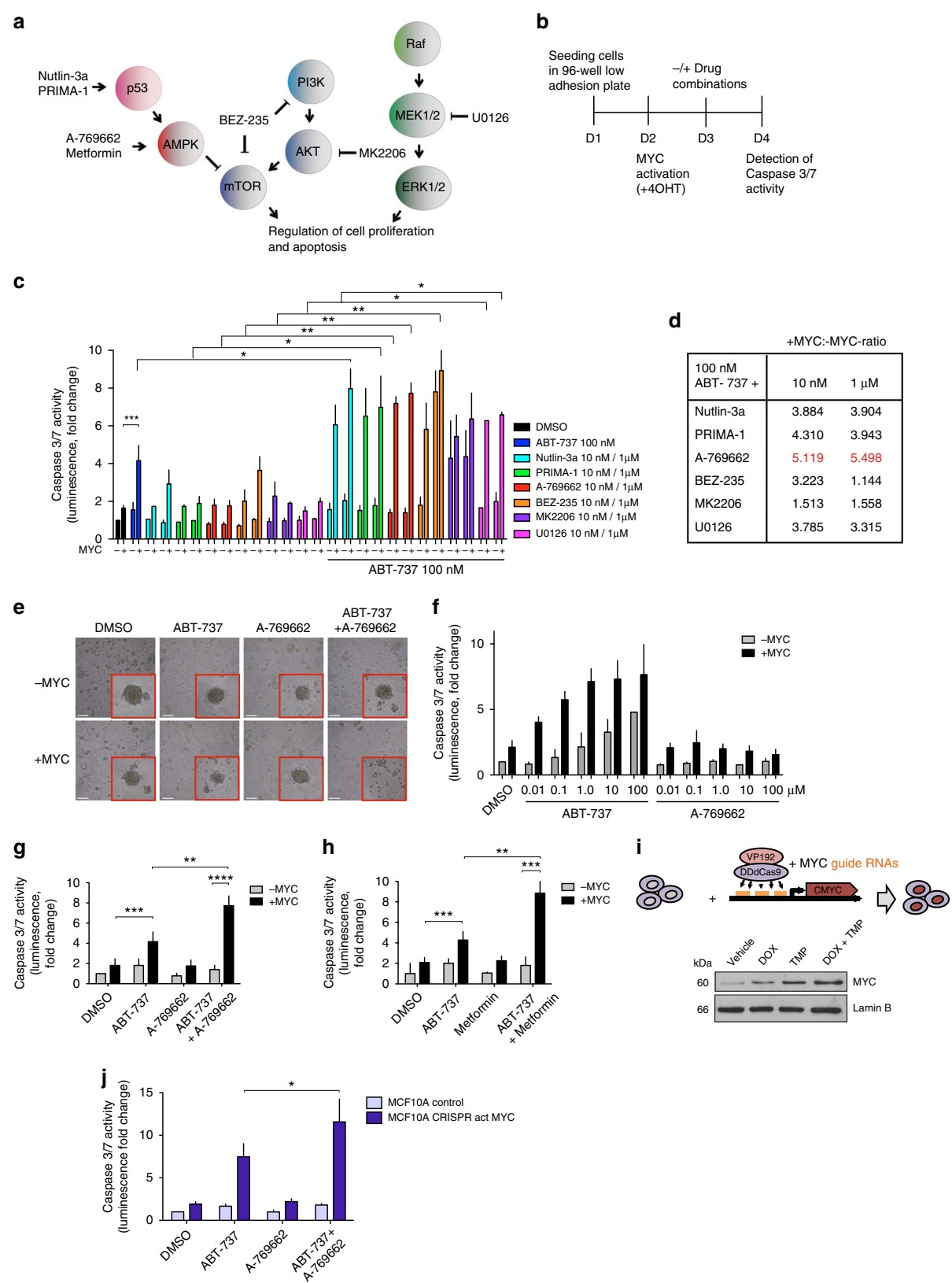

observed in the non-cancerous samples (Supplementary Figure 3D). While ABT-737 alone occasionally promoted apoptosis, the most prominent responses were obtained with the combination (Fig. 4e–f). Among the combination-treated cultures, the strongest apoptotic response was observed in PDEcs derived from

MYC-high tumors and the overall difference in apoptosis between MYC-high and MYC-low PDEcs was statistically significant (Fig. 4f). Aside from the MYC link, examination of the samples' clinical annotations revealed no other significant correlations with strong apoptotic responses (Supplementary

**Fig. 2** AMPK activation potentiates MYC-dependent apoptosis by ABT-737. **a** Drug-targeted pathways. **b** Protocol for drug combination testing. MCF10A MycER cells were allowed to form mammospheres for 24 h. MYC was activated with 100 nM 4OHT for 24 h followed by 24 h incubation with drug combinations. **c** Combination drug testing to identify pharmacological triggers of MYC-dependent apoptosis. The drugs were administered as single agents or as ABT-737 with and without MYC activation. Each drug was tested in two concentrations. $N = 3$ biological repeats. Student's $t$-test (unpaired), SD. **d** The relative level of apoptosis with and without MYC activity ($+$MYC: $-$MYC ratio). The ratio was calculated from fold-change in **c**. **e** Representative images of drug-treated mammospheres. **f**, **g** Sensitization to MYC-dependent apoptosis by 100 nM ABT-737 with either 1 μM A-769662 or 10 mM metformin. Mammospheres were treated as in **b**. Student's $t$-test (unpaired), $N = 3$ biological replicates, SD. **h** Activation of AMPK alone does not sensitize to MYC-dependent apoptosis. $N = 3$ biological repeats, SD. **i** CRISPR/dead-Cas9-mediated transcriptional activation of MYC. HEK293 and MCF10A cells were transduced with vectors encoding dCas9-VP192 transcription-activating construct and MYC-promoter-targeted guide-RNAs. Western blot analysis shows MYC expression levels after 72 h treatment with doxycycline (DOX) and trimethoprim (TMP). Lamin B: Loading control. **j** CRISPR-mediated induction of endogenous MYC sensitizes cells to apoptosis by ABT-737+A-769662. Mammospheres with and without dCas9-VP192+MYC-gRNA were treated and analyzed as in **b**. Student's $t$-test (unpaired), $N = 3$ biological replicates, SD

---

Table 1). Consistent with our cell culture findings, A-769662-induced AMPK activation and BIM upregulation in PDEcs (Fig. 4g).

**Antitumor effects of metformin and BCL-2/-X$_L$ inhibition.** Based on the strong signals of activity in PDEcs, we went on to explore the antitumor activity of combined AMPK activation and BCL-2/BCL-X$_L$ inhibition in vivo. For in vivo testing, we formulated an "AB" (A: AMPK activator, B: Inhibitor of anti-apoptotic BCL-2 proteins) regimen. For the "B" component, we used either ABT-263/navitoclax (ABn) or ABT-199/venetoclax (ABv). Navitoclax is an orally bioavailable analogue of ABT-737 that also inhibits both BCL-2 and BCL-X$_L$, whereas venetoclax is a distinct, oral agent that selectively inhibits BCL-2. For the "A" component, metformin was implemented in place of A-769662. Metformin is an AMPK-activating agent with a long history as a type II diabetes drug that has well-established adverse-effect and safety profiles[28]. The analysis of four different BH3 mimetics in combination with either A-769662 or metformin confirmed that metformin has similar apoptosis-potentiating action as A-769662 (Fig. 5a, Supplementary Figure 4A). Furthermore, metformin in combination with navitoclax (ABn) showed similar pro-apoptotic activity in MYC-high PDEcs (Fig. 5b, compare with Fig. 4e).

TNBC is a breast cancer subtype with poor clinical outcome and commonly elevated MYC levels[29]. In a panel of 16 TNBC cell lines the ABn treatment strongly reduced viability in cell lines with elevated MYC expression. At the low apoptosis-inducing concentration of navitoclax (EC$_{20}$), combination with metformin potentiated apoptosis in 11 out of 11 MYC-positive TNBC cell lines, whereas no such effect was seen in 4 out of 5 cell lines with low or undetectable MYC expression (Fig. 5c–e, Supplementary Figure 4B–C). These experiments confirmed the MYC-associated apoptotic potential of the AB regimen, supporting its use for in vivo studies.

To evaluate the antitumor potential of AB treatment in vivo, we used a patient-derived xenograft (PDX) model based on graftable samples from a metastatic TNBC. The mammary gland-grafted TNBC sample formed aggressive tumors that invaded through the peritoneum and formed multiple metastases in the abdominal cavity (Fig. 5f). Sequencing of the sample revealed typical TNBC-associated mutations such as p53 R248W (Supplementary Figure 5A). Passaging in vivo allowed us to generate mouse cohorts for multiple treatment arms and high MYC expression was verified in passaged tumors (Fig. 5g). TNBC samples were transplanted into mammary fat pads and treatments were initiated 2 weeks post operation. Although navitoclax or metformin failed to show activity as single agents, the ABn combination treatment strongly inhibited tumor growth (Fig. 5h). In addition, ABn treatment significantly

extended the survival of TNBC-grafted mice. Taken together, the combination of metformin and navitoclax exerts MYC-associated antitumor activity capable of prolonging the survival of mice transplanted with a highly aggressive and invasive TNBC specimen.

**ABn does not benefit mice with Myc-low mammary tumors.** To address the importance of high MYC expression for treatment efficacy, we employed two mouse models of breast cancer with low MYC expression; the MMTV-PyMT model with tumors syngrafted to FVB mice and an ER+ PDX model with tumors grafted to NSG mice. Mice with orthotopically engrafted tumors were subjected to the ABn as before. In contrast to FVB WapMYC syngrafts, the FVB MMTV-PyMT tumors did not respond to navitoclax or metformin, either as single agents or in combination (Supplementary Figure 5E). Although tumors in the MYC-low ER+ PDX model responded well to navitoclax as a single agent (we note that BCL-2 expression is often reported as enriched in this breast cancer subtype[30]), combining it with metformin did not offer any added benefit (Supplementary Figure 5F). These results are consistent with the notion that high MYC expression defines tumors that are most likely to respond to ABn treatment.

**ABn activates MYC-dependent immunogenic apoptosis in vivo.** Accumulating evidence indicates that the therapeutic effects of current anticancer drugs, including chemotherapy, arise not only from direct tumor cell killing, but also from enhanced immunoactivation[31–33]. The immunoactivating signals may derive from the injured tumor, for example, in the form of liberated pro-inflammatory cytokines[32] or via release of tumor antigens and neoantigens[31]. Notably, some forms of drug-induced cell death appear more immunogenic than others[34].

To explore whether the ABn treatment promotes immunoactivation, we generated cohorts of immunocompetent WapMyc tumor syngrafted mice, which were treated with single drugs or combination (Fig. 6a). In mice, 250 mg/kg/d dosing of metformin achieves similar drug plasma concentrations as measured for diabetic patients treated with 2 g/day[35,36]. The metformin safety profile allowed dose escalation from the standard antidiabetic dosage to maximize the apoptotic effects. Thus, for antitumor treatments the mouse cohorts were treated with the standard antidiabetic dose of 300 mg/kg/d or with an escalated 600 mg/kg/d dose. A set of tumors dissected immediately after the treatments and before the follow-up (on-treatment samples) exhibited striking AMPK activation, BIM upregulation, and apoptosis in response to ABn already with the antidiabetic dose of 300 mg/kg/d (Fig. 6b, Fig. 6c, Supplementary Figure 5B). The combination treatment triggered notable regions of dying cells, or "apoptotic ponds" (Fig. 6c), an effect rarely seen in vivo

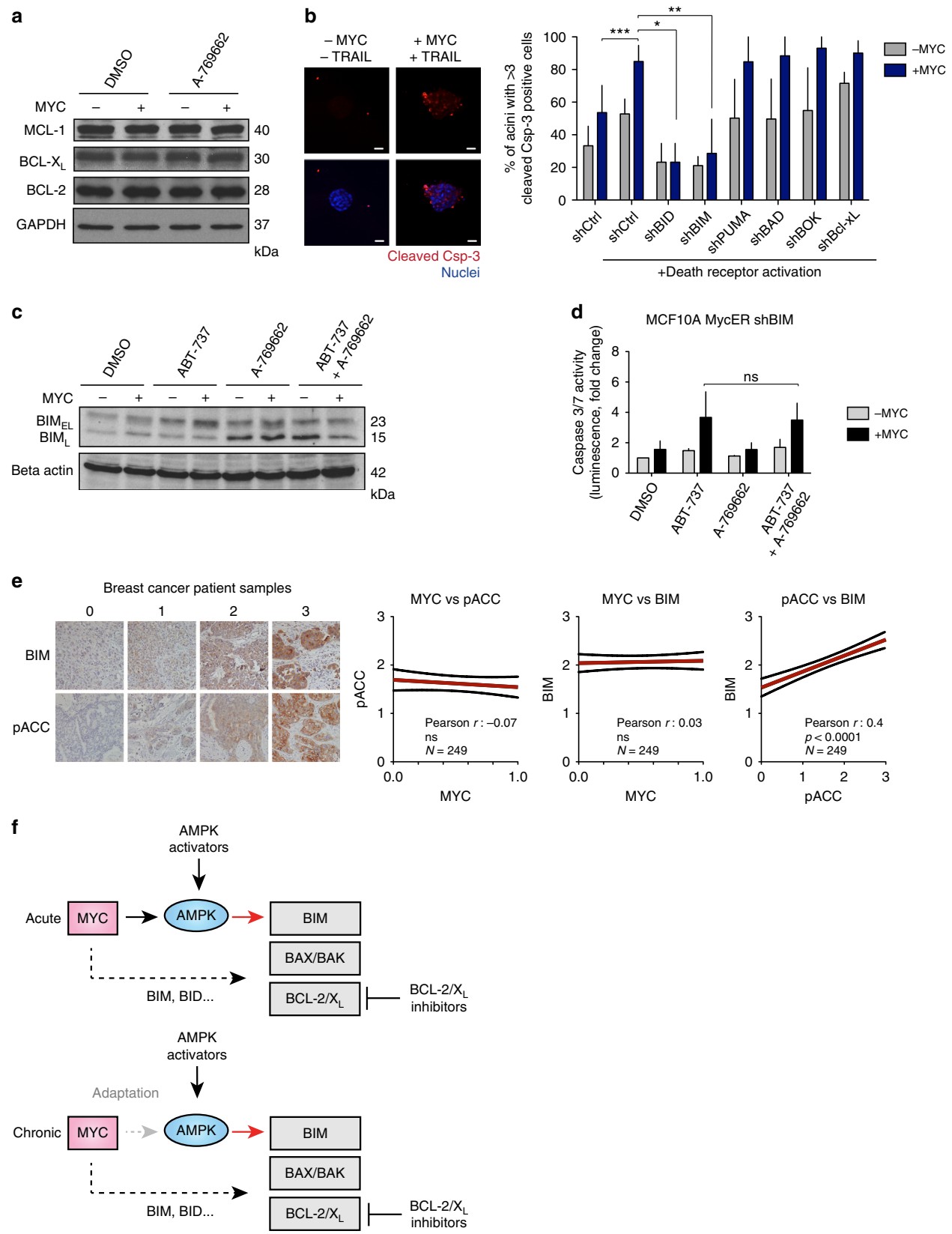

due to the rapid clearance of debris by neighboring cells and macrophages.

Immunohistochemical analysis of the ABn on-treatment tumor samples revealed significant enrichment of CD4+ T helper cells and CD8+ cytotoxic T cells (Fig. 6d). Recent investigations have highlighted the critical role of systemic immune response in eradication of tumors by immunotherapies[37]. Examination of the peripheral blood of ABn-treated mice showed enrichment of

**Fig. 3** AMPK activation upregulates BIM to sensitize cells to apoptosis. **a** AMPK activation does not alter the expression of anti-apoptotic BCL-2 family proteins. MCF10A MycER cells were treated with vehicle or 1 μM A-769662. MYC was activated with 100 nM 4OHT for 24 h before the 24 h drug treatment. **b** shRNA screen to determine pro-apoptotic regulators of MYC-dependent apoptosis. Left: Apoptosis induced by MYC and TRAIL in 3D MCF10A acini. Right: shRNA-transduced cells were seeded into Matrigel to form acini and MycER was activated on day 20. The 3D acini were treated with 100 ng/ml of TRAIL for 72 h. Apoptosis was scored using active caspase-3 readout. Student's $t$-test (unpaired), $N = 3$ biological replicates, SD. **c** The effect of A-769662 on $BIM_{EL}$ and $BIM_L$ expression. MYC was activated in MCF10A MycER for 24 h followed by 24 h treatment with 100 nM ABT-737, 1 μM A-769662, or with combination. Lamin B: Loading control. **d** BIM knockdown blunts AMPK-mediated sensitization to apoptosis. MCF10A MycER shBIM cells were treated as in **c** followed by quantitative analysis of apoptosis. Student's $t$-test (unpaired), $N = 3$ biological replicates, SD. **e** Correlations between nuclear MYC protein and BIM or AMPK activity (pACC) in breast cancer. Left: Representative images of BIM and pACC staining intensity groupings. Right: High AMPK activity associates with elevated BIM expression in breast cancer. Blinded scoring of BIM and pACC levels, with intensities from 1 to 3 in a breast cancer tissue microarray. **f** A model for AMPK-induced reactivation of MYC-dependent apoptosis in cancer. Acute: Acute activation of MYC sensitizes cells to apoptosis via BIM-dependent modulation of the interactions between pro-apoptotic BAK/BAX and the anti-apoptotic BCL-2/$X_L$ proteins. Likely, other BH3-only proteins also contribute to the apoptotic sensitization (dotted arrow). Pharmacological activation of AMPK enhances the BIM load in MYC-overexpressing cells (red arrow), which together with inhibition of BCL-2/$X_L$ triggers apoptosis. Chronic: Tumors may escape AMPK-BIM pathway via metabolic adaptation or by acquiring resistance mechanisms (thick dotted arrow), but the pro-apoptic pathway is still amenable to reactivation by pharmacological AMPK activators

CD4+ T and NK cells (Fig. 6e), thus suggesting not only local but also systemic immune activation.

**ABn induces T lymphocyte infiltration and exhaustion**. In the experiments with WapMyc tumor syngrafted mice, ABn treatment significantly inhibited tumor growth in a dose-dependent manner with no observable adverse effects (Fig. 6f, Supplementary Figure 5C–D). Importantly, the combination treatment also prolonged survival. At the end of the follow-up period, half of the mice that received ABn with the antidiabetic metformin dose were still alive, in contrast to all other cohorts, which had no survivors (Fig. 6g). The higher 600 mg/kg/d dose of metformin together with navitoclax provided a striking survival benefit, with 75% of the ABn-treated mice surviving over the study period (Fig. 6h). A survival benefit was also observed in the cohort receiving the high metformin dose alone, suggesting that high metformin concentrations may have other therapeutic effects besides induction of apoptosis.

Although, the ABn treatment controlled tumor growth (Fig. 6f), it failed to prevent post-treatment tumor recurrence. To explore why the tumors regrew despite signs of immunoactivation, we analyzed immune infiltrates in the post-treatment tumors. Tumors of equivalent size (Ø 2 cm) were isolated after the treatments and equivalent amounts of tumor mass were homogenized into single cell suspension for flow cytometric analysis with panels of immunophenotype-specific antibodies. The analyses revealed post-treatment persistence and activity of tumor-infiltrating NK cells (CD3−/NK1.1+), NKT cells (CD3+/NK1.1+) and CD4+ T cells, and a general enrichment of total leukocytes in the treated tumors (Fig. 6i). However, the number of active cytotoxic T cells (CD8+/CD107+) was lower in the treated tumors in comparison to controls, whereas the opposite was true for interferon-gamma (IFN-γ)-secreting cytotoxic T cells, which were observed in higher levels after ABn treatment (Fig. 6i). IFN-γ secretion by T cells is necessary for T cell activity, but also promotes T-cell exhaustion and adaptive immune resistance by upregulating inhibitory PD-1/PD-L1 signaling[38,39]. PD-1 is an inhibitory coreceptor found on T lymphocytes that is responsible for feedback inhibition of T-cell activation[34]. We observed increased ratio of PD-1-positive cytotoxic T cells in the tumors that had been treated with ABn, suggesting the onset of T-cell exhaustion (Fig. 6j). To summarize, ABn treatment leads to increased infiltration of tumors by lymphocytes of both adaptive and innate immunity, with markers of cytolytic activity and T-cell exhaustion.

**ABv with anti-PD-1 provides durable antitumor response**. T-cell exhaustion can occur due to persistent antigen stimulation in viral infections and is also a key target of immunotherapies for cancer. Antibodies targeting the PD-1 inhibitory receptor pathway have demonstrated significant antitumor activity and have been approved for use in cancer patients[31]. The observation of T-cell exhaustion markers post-ABn treatment suggested that more durable treatment responses could be achieved by combining AB with anti-PD-1 therapy. For these experiments, new cohorts of WapMyc-syngrafted mice were generated and treated with ABv, which includes the BCL-2-specific inhibitor venetoclax (ABT-199). Venetoclax was chosen here for its translational potential, as it was recently granted accelerated approval by the FDA for a cancer indication[40] and it also synergized with metformin to induce MYC-dependent apoptosis (Fig. 5a, Supplementary Figure 4A).

We hypothesized that AB-induced apoptosis is crucial for triggering the initial antitumor immunity that is subsequently blunted by PD-1-dependent immunosuppression. Accordingly, we designed a preclinical treatment protocol in which ABv was first administered as a neoadjuvant (pre-surgery) treatment. To avoid early killing of control mice and to extend the experimental time-frame, the tumors were surgically resected at the end of the first treatment period (Fig. 7a). Following surgery, the mice received as an adjuvant (post surgery) treatment single drugs or ABv, either in combination with control IgG or anti-PD-1 antibody. Neoadjuvant ABv significantly inhibited tumor growth, although the effect was weaker than in previous studies with ABn (Fig. 7b). After tumor resection, all control mice developed aggressive secondary tumors in less than 1 week, and had to be killed within 2 weeks after resection (Fig. 7c–d, see Fig. 7g for survival). Anti-PD-1 as an adjuvant did not provide any treatment benefit, suggesting that these tumors were not immunotherapy responsive by default (Fig. 7c–d). However, the ABv treatment (with or without anti-PD-1) efficiently prevented secondary tumor formation in the adjuvant setting (Fig. 7c). Strikingly, after completion of the adjuvant treatment, the ABv+IgG-treated mice soon started to develop tumors; whereas, the ABv+anti-PD-1-treated mice remained essentially tumor free (Fig. 7d–e). Only one mouse in the latter group developed a tumor in one flank (Fig. 7e). Notably, mice that received adjuvant venetoclax+anti-PD-1 treatment also showed reduced tumor re-growth potential, suggesting that even milder apoptosis induction by venetoclax alone could render tumors susceptible to immunotherapy. All tumors were immunoprofiled after adjuvant treatment, and the lone tumor that appeared in the ABv+anti-PD-1 cohort was biopsied. Whereas the

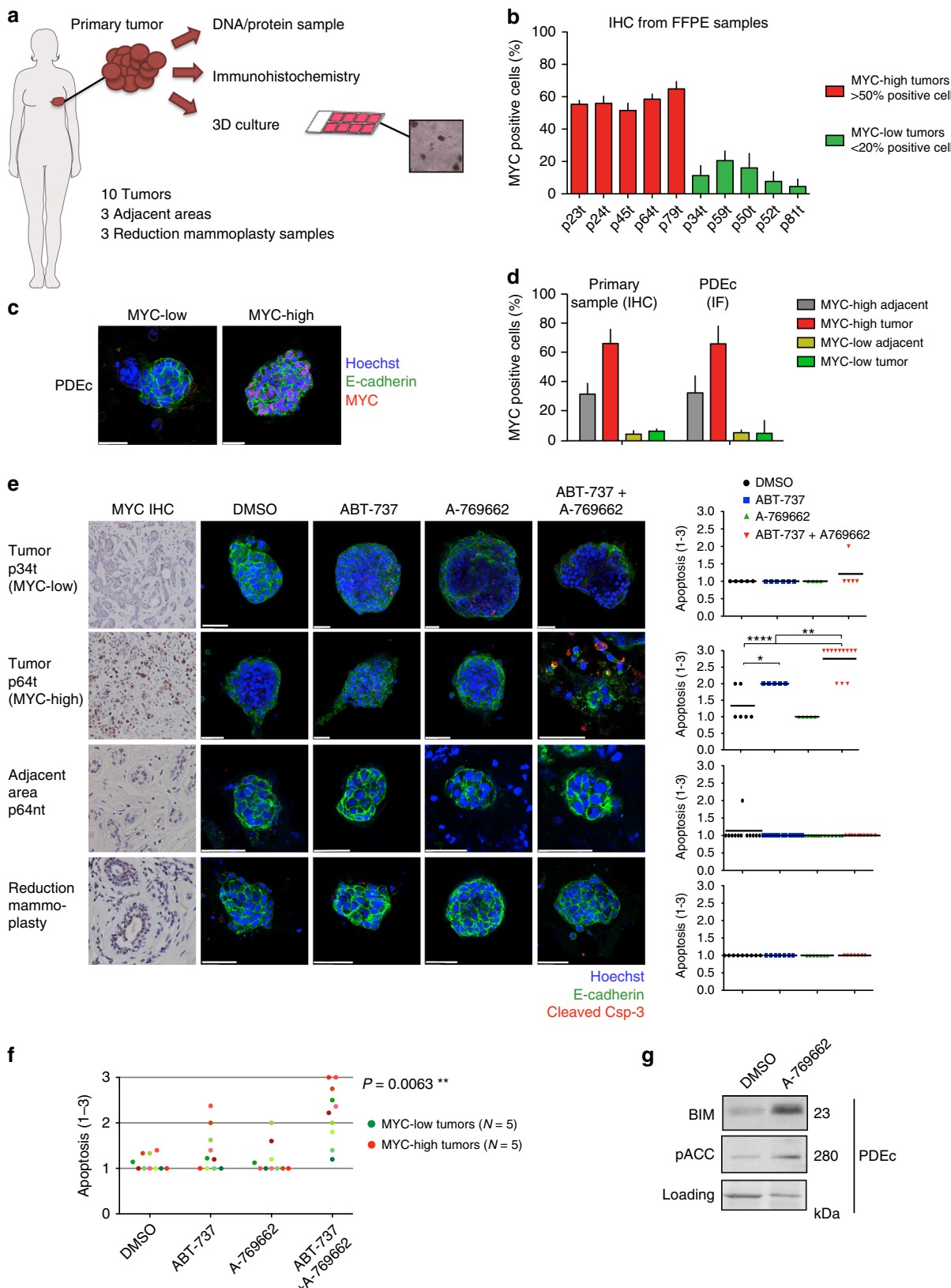

ABv+IgG-treated tumors displayed reduction in active CD8+ cytotoxic T cells and minor changes in other lymphocyte subsets, the biopsied sample from ABv+anti-PD-1-treated tumor showed a general increase in the number of TILs and activity markers (Fig. 7f). The ABv+anti-PD-1 treatment offered striking survival benefit, with all the mice in the group surviving through the 30-day follow-up period (Fig. 7g).

Next, we investigated the durability of the ABv+anti-PD-1 treatment response, extending the follow-up period to 60 days, and also assessed its efficacy in comparison to the combination of

**Fig. 4** AMPK activation combined with BCL-2/X_L inhibition induces MYC-associated apoptosis in breast cancer patient-derived explant cultures. **a** Workflow for breast cancer patient tissue-derived explant cultures (PDEc). Tumor pieces were brought to the lab directly after surgery and cut into three pieces; one for DNA/protein extraction, one for immunohistochemical (IHC) analyses and one for 3D culture. **b** IHC analysis of MYC in breast tumors. Paraffin-embedded samples were immunostained for MYC and ≥5 fields of view (FOV) from 1 to 3 sections scored. MYC-high tumors: >50% positive cells/FOV, MYC-low tumors: <20%-positive cells/FOV. N = 10 tumors. **c** Representative images of MYC-low and MYC-high PDEc cultures. The PDEc samples were cultured for 7 days, stained and imaged by confocal microscopy (IF). **d** MYC status before and after 3D culture. MYC positivity scored as in **b**. **e** ABT-737+A-769662-induced apoptosis in MYC-high, MYC-low, and non-cancerous (tumor adjacent tissue and reduction mammoplasty) PDEc cultures. Samples were adjusted to culture for 6 days and treated for 24 h with vehicle (DMSO), 1 μM ABT-737, 10 μM A-769662 or ABT-737+A-769662 combination. The level of apoptosis was scored from confocal immunofluorescence images. The left panel shows representative images. The graphs at right plot the quantification of apoptosis. Apoptosis was scored as 1 = <10% apoptotic cells/explant, 2 = >10% apoptotic cells/explant in a cohesive structure, 3 = >10% apoptotic cells/explant in a deteriorated structure. Each dot indicates one fragment. Horizontal lines: Average, Student's *t*-test (unpaired). **f** Summary of the apoptotic response in PDEc. Student's *t*-test (unpaired). The different shades of green represent the different MYC-low tumor samples, and the different shades of red represent the different MYC-high tumor samples. **g** AMPK activation upregulates BIM in PDEc culture. pACC was used as an AMPK activity marker. The sample was treated with DMSO or 10 μM A-769662 for 24 h

paclitaxel and anti-PD-1 antibody. Taxanes (including paclitaxel) are standard of care in the treatment of patients with breast cancer and encouraging results were reported recently from the Phase III IMpassion130 study, which included combination of nab-paclitaxel plus the anti-PD-L1 antibody atezolizumab for patients with advanced TNBC[41]. We noted that both ABv and paclitaxel treatment caused low white blood cell counts (Fig. 7h); however, only the paclitaxel+anti-PD-1 combination caused significant elevation of alanine transaminase (ALAT), a marker of liver injury (Fig. 7i). Tumor growth was clearly restricted in cohorts that received either ABv+anti-PD-1 or paclitaxel+anti-PD-1 (Fig. 7j). Comparing the treatments, the therapeutic benefit was similar in both tumor growth and survival analyses (Fig. 7j–k). One cohort of ABv+anti-PD-1-treated mice was divided into two smaller cohorts - one receiving three cycles of ABv+anti-PD-1 and the other one cycle as in the previous experiment. Only one mouse survived to the end of the 60-day follow-up period, and it had received three cycles of ABv+anti-PD-1 (Fig. 7k). Although ABv+anti-PD-1 treatment does not cure mice syngrafted with aggressive tumors, these data indicate that it significantly delays tumor re-growth and markedly improves their survival.

In conclusion, we propose that the combination of cell death-inducing AB therapy together with anti-PD-1 immunotherapy provides a potent tumor-targeting strategy with high translational potential in tumor types with high MYC expression.

## Discussion

MYC lacks the classical drug-binding sites of typical enzymes; therefore, most therapeutic strategies have focused on inhibiting its expression, stability or interaction with DNA binding partners[42]. However, such approaches may come with unwanted side effects, such as reduced sensitivity to front-line chemotherapies[43]. Alternative, synthetic-lethal MYC-targeting (SL-MYC) approaches aim to exploit MYC-dependent cancer vulnerabilities without directly inhibiting MYC. For example, pharmacological targeting of the spliceosome[44], RhoA and SRF survival signaling[45,46], fatty acid oxidation[47], or glutamine metabolism[12,48] induces MYC-selective apoptosis. Also MYC-induced AMPK generates a cancer cell vulnerability via AMPK-induced accumulation of mitochondrial p53, which binds to and stimulates BAK activation[22]. While the concept of SL-MYC-based cancer therapy is alluring based on MYC's overexpression across many cancer types, most currently available pharmacological or biological agents with proven SL-MYC activity are not suitable for testing in the clinic.

Here, we have established the strong pro-apoptotic and anti-tumor potential of AMPK-activating molecules ex vivo and in vivo. The pro-apoptotic activity of AMPK presents itself in a specific context of high MYC expression and lowered BCL-2/BCL-X_L activity. The pro-apoptotic action of AMPK is coupled to induction of the BH3-only protein BIM. BIM contains one of the most potent BH3 death domains in the BCL-2 family, with a capacity to engage all anti-apoptotic proteins with high affinity and to activate death effectors, such as BAX[49,50]. Once activated, BAX and BAK form pores in mitochondria to release cytochrome *c* and other pro-apoptotic mediators into the cytosol[51]. While earlier studies have shown that classical cell damage-associated stress signaling and endoplasmic reticulum stress upregulates BIM[52], our results extend these findings to AMPK-dependent stress signaling. In summary, pharmacological activation of AMPK provides a tumor cell selective but otherwise non-damaging route for induction of therapeutic apoptosis as a safer alternative to more toxic agents. In our breast cancer TMA set high MYC status did not correlate with high BIM expression or high AMPK activity; however, high AMPK activity correlated with high BIM expression. Thus, chronically elevated MYC levels may not be able to sustain pro-apoptotic levels of AMPK-BIM activity long-term and it is tempting to speculate that pharmacological stimulation of AMPK reactivates the MYC-associated apoptotic machinery intrinsic to tumor cells (Fig. 3f). However, the exact mechanistic role of MYC as a sensitizer of AMPK-mediated tumor apoptosis remains to be clarified.

Our studies with an orally bioavailable ABn regimen (metformin+navitoclax) consistently demonstrated therapeutic potential and efficacy in different models of MYC-overexpressing breast cancer, including TNBC cell lines, breast cancer explants, the WapMyc mouse model of breast adenocarcinoma, and patient-derived xenografts. Notably, no therapeutic potential was observed for the ABn regimen in two breast cancer models with low MYC expression status. Metformin is an inexpensive insulin-lowering drug with an excellent safety profile, which has made it the most prescribed drug for type II diabetes. In addition to metformin's well-known antidiabetic activities, many epidemiological and retrospective studies have suggested antitumoral effects in cancers including breast cancer[53,54]. Metformin is an antihyperglycaemic drug, which inhibits hepatic gluconeogenesis in vivo through a LKB1/AMPK-dependent mechanism and via inhibition of mitochondrial respiratory chain complex I[55]. With regard to cancer, preclinical studies have suggested that the tumor-preventing action of metformin is mediated through activation of the AMPK pathway, which leads to suppression of metabolic ATP-consuming processes such as glycolysis and fatty acid oxidation[54]. These effects of metformin on cell energetics are likely to impact tumor growth by mechanisms unrelated to apoptosis and, indeed, we observed notable antitumoral effects in the syngraft WapMyc model when metformin was administered at a dose higher than the antidiabetic dose. However, we could

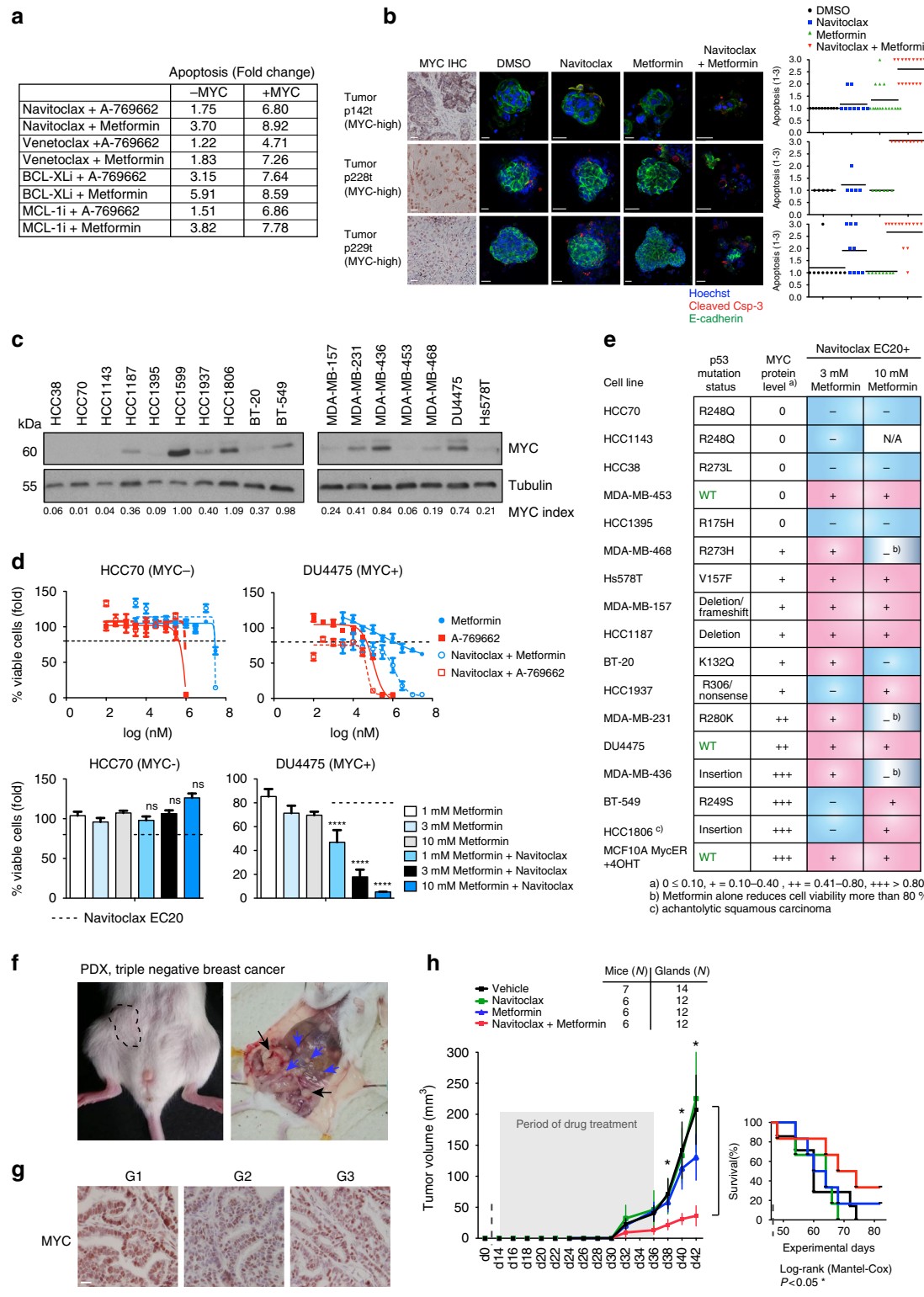

**a** Apoptosis (Fold change)

|  | −MYC | +MYC |
|---|---|---|
| Navitoclax + A-769662 | 1.75 | 6.80 |
| Navitoclax + Metformin | 3.70 | 8.92 |
| Venetoclax + A-769662 | 1.22 | 4.71 |
| Venetoclax + Metformin | 1.83 | 7.26 |
| BCL-XLi + A-769662 | 3.15 | 7.64 |
| BCL-XLi + Metformin | 5.91 | 8.59 |
| MCL-1i + A-769662 | 1.51 | 6.86 |
| MCL-1i + Metformin | 3.82 | 7.78 |

**e**

| Cell line | p53 mutation status | MYC protein level a) | Navitoclax EC20+ 3 mM Metformin | Navitoclax EC20+ 10 mM Metformin |
|---|---|---|---|---|
| HCC70 | R248Q | 0 | − | − |
| HCC1143 | R248Q | 0 | − | N/A |
| HCC38 | R273L | 0 | − | − |
| MDA-MB-453 | WT | 0 | + | + |
| HCC1395 | R175H | 0 | − | − |
| MDA-MB-468 | R273H | + | + | − b) |
| Hs578T | V157F | + | + | + |
| MDA-MB-157 | Deletion/frameshift | + | + | + |
| HCC1187 | Deletion | + | + | + |
| BT-20 | K132Q | + | + | − |
| HCC1937 | R306/nonsense | + | − | + |
| MDA-MB-231 | R280K | ++ | + | − b) |
| DU4475 | WT | ++ | + | + |
| MDA-MB-436 | Insertion | +++ | + | − b) |
| BT-549 | R249S | +++ | − | + |
| HCC1806 c) | Insertion | +++ | − | + |
| MCF10A MycER +4OHT | WT | +++ | + | + |

a) 0 ≤ 0.10, + = 0.10–0.40, ++ = 0.41–0.80, +++ > 0.80
b) Metformin alone reduces cell viability more than 80 %
c) achantolytic squamous carcinoma

not observe any apoptotic effects when metformin was administered alone. Therefore, metformin alone may not trigger apoptosis, but has pro-apoptotic effects, via BIM upregulation, which can be exploited therapeutically using BH3 mimetics.

We show that the strong induction of apoptosis by ABv treatment (metformin+venetoclax) increases the density of tumor-infiltrating lymphocytes (TILs). This effect was also observed in tumors that developed after AB treatment, as indicated by increased densities of NK and NKT cells expressing the CD107+

cytolytic marker. However, CD8+ T cells were decreased in the post-treatment tumors and the remaining cells exhibited markers of T-cell exhaustion[56]. These findings provided a rationale for modified ABv therapy, which included an anti-PD-1 antibody as the third component. In experiments using a revised ABv treatment protocol, apoptosis was induced by ABv in a neoadjuvant setting. After removal of the tumors, the treatment was continued in the adjuvant setting using ABv together with anti-PD-1 antibody. Among all the treatment designs tested in the present study,

**Fig. 5** ABn treatment reduces viability of MYC-high triple-negative breast cancer cell lines, inhibits tumor growth, and extends survival in patient-derived xenografts. **a** A summary of the results of BH3 mimetic screen in MCF10A MycER cells. The numbers refer to fold change as in Supplementary Figure 4A. The drugs were navitoclax, venetoclax, a BCL-X$_L$-selective inhibitor A-1155463 and an MCL-1-selective inhibitor A-1210477. **b** AB treatment-induces apoptosis in MYC-high PDEc. The cultures were treated with DMSO, 1 μM navitoclax, 10 mM metformin or combination for 24 h. Blinded scoring of apoptosis was carried out for all samples. **c** MYC expression in 17 triple-negative breast cancer cell lines. MYC index: MYC intensity normalized to a blot-to-blot reference sample, highest intensity band (HCC1599) and loading control. N.B. HCC1599 excluded from the final analysis due to poor growth. Tubulin: Loading control. **d** The effect of AB treatment on the viability of MYC-low and MYC-high TNBC cell lines. The upper panel shows kill curves and the dashed line marks the EC$_{20}$ of navitoclax (20% reduction in survival); see also Fig.S4E. The lower panel shows metformin effect at EC$_{20}$ of navitoclax. Student's *t*-test (unpaired), SD. **e** Summary table of drug treatments in TNBC cell lines. The cell lines were categorized according to MYC index (0 = undetectable MYC expression; + to +++ = relative MYC expression level). Blue boxes indicate statistically non-significant and pink boxes statistically significant differences between each single-agent treatment and the corresponding combination. Student's *t*-test (unpaired). **f** Representative images of tumors developing in TNBC-PDX mice. Black arrows: Primary tumors developing at the site of the tumor grafts; blue arrows: Metastases. **g** TNBC-PDX tumors retain MYC expression during in vivo passaging. Tumor generations G1–G3. **h** Effect of AB treatment on tumor growth and survival in cohorts of TNBC-PDX mice. The mice were treated with vehicle, 100 mg/kg/d navitoclax, 600 mg/kg/d metformin or the combination for 21 days, and followed up until day 60. Student's *t*-test (unpaired), SEM. In the survival graph: *P*-value: Significant difference between vehicle and AB-treated cohorts

only the ABv+anti-PD-1 regimen prevented all tumor-related deaths during the 30-day experimental period. Only a single tumor developed in one mouse of the ABv+anti-PD-1 group during the post-treatment follow-up period and a biopsy of this tumor showed enhanced densities of tumor-infiltrating lymphocytes. In 60-day follow-up the tumors eventually reappeared; nevertheless, ABv+anti-PD-1 had similar or slightly enhanced antitumor activity compared to paclitaxel+anti-PD-1, and it also showed less signs of liver toxicity measured by plasma ALAT levels.

The durable efficacy of the ABv+anti-PD-1 treatment would be consistent with a model, suggested by Sharma & Allison[31] that targeted therapies with strong apoptotic actions serve as "cancer vaccines" by liberating tumor antigens and neoantigens from the dying cancer cells. In this scenario, the role of checkpoint inhibitors is to permit sustained T-cell responses. In support of this notion, administration of anti-PD-1 antibody alone did not inhibit tumor growth at all but showed remarkable efficacy when administered together with ABv treatment.

In summary, we report here the discovery and preclinical validation of a clinically applicable synthetic-lethal MYC-targeted therapeutic strategy for the treatment of breast cancer. The ABv+anti-PD-1 combination showed remarkable efficacy in a number of breast cancer models, which was somewhat surprising in light of the fact that BCL-2 dependency is most often observed in hematologic malignancies. The applicability of the ABv+anti-PD-1 combination for treating other cancer types with high MYC and BCL-2/BCL-X$_L$ status is currently being investigated in preclinical models.

## Methods
**Transgenic animals.** All animal experiments were approved by the National Animal Ethics Committee of Finland (License number: ESAVI-3216/04.10.07/2013), and the mouse colonies were maintained according to the protocols of the Experimental Animal Committee of the University of Helsinki. FVB and WapMyc mice (FVB.Cg-Tg(WapMyc)212Bri/J) were obtained from the Jackson Laboratory. To activate Wap-promoter for tumorigenesis, WapMyc females of over 8-weeks of age underwent two pregnancies. The drug treatments were started when the tumor size reached Ø 5 mm. For genotyping, DNA was extracted from tail samples with DNAreleasy Kit (NIPPON Genetics EUROPE GmbH) according to the manufacturer's protocol. The genotype was analyzed by PCR using the following primers: IMR0135 Fwd 5′-CATCCCTGTGACCCCTCC-3′ and IMR0136 Rev 5′-CTCCAAACCACCCCCCTC-3′.

**Patient-derived xenografts.** The patient breast cancer samples for grafting were received from Oncotest (now part of Charles River) and expanded once in NSG mice (Charles River).

**Mammary tumor transplantations.** WapMyc tumors: For the procedure, 3-week-old female FVB-recipient mice were given painkillers and anesthetized using inhaled 2.5% isoflurane. Subsequently, the anterior part of 4$^{th}$ mammary gland was cleared from the endogenous epithelium. 10$^5$ primary WapMyc tumor cells were

injected into the cleared gland in 10 μl of PBS supplemented with 5% FCS and the wound was sealed with clips. The mice were given analgesics immediately after the transplantation and monitored after the surgery. Drug treatment was started 1 week after the transplantations and the mice were treated and followed up as detailed in the text. In the adjuvant treatment setting, the tumors were surgically removed from the mice under isoflurane anesthesia, and the adjuvant treatment was started after 3 days of recovery. Animals which formed tumors in the surgery scar were excluded from the experiment.

MMTV-PyMT tumors: MMTV-PyMT tumor cells (kind gift from Dr. Pauliina Kallio and Kari Alitalo, University of Helsinki) were syngrafted to FVB hosts similarly as described in FVB WapMyc syngrafts. The tumor cells were isolated from MMTV-PyMT donor and 10$^6$ cells were transplanted per gland.

PDX tumors: TNBC-PDX transplantations were performed on immunocompromised NOD scid gamma (NSG, NOD.Cg-Prkcd$^{scid}$I12rg$^{tm1Wjl}$/SzJ) obtained from Charles River. For ER+PDX, 60-day 17β estradiol pellets (Innovative Research America) were inserted into the neck of the animals to promote tumor growth. Mice were killed when the tumor reached Ø 2 cm. Drug treatments were started 14 days after the transplantation. Tumor growth was monitored every second day.

**Statistical modeling of tumor growth.** The mice were randomized into treatment groups and the tumor measurements were obtained with electronic caliper. The results were averaged from two measurements performed by two individual researchers. The formula for calculating tumor volume was $V = d^2 \times D/2$, where $d$ = the shortest diameter and $D$ = the longest. For statistical modeling of tumor growth and treatment effect in the animal experiments, we used a mixed-effects modeling framework that makes use of the whole longitudinal growth profile in assessing the treatment and other fixed effects, while the random effects account for individual tumor and animal-specific variation in growth patterns[57]. Moreover, the categorizing mixed-effects model takes into account both the growing and stable or poorly growing tumor sub-categories when testing treatment effects through model parameters, such as tumor growth rates (slopes) or overall tumor levels (offset).

**Drug treatments in mice.** ABT-737: The mice were treated with either 100 mg/kg/d ABT-737 solution or an equal volume of carrier solution administered via intraperitoneal injections (i.p.) for 21 consecutive days. ABT-737 was received from Abbott (now AbbVie Inc). After the treatment the mice were followed up for 21 days. Navitoclax, venetoclax, metformin: The mice were treated with vehicle (5% EtOH, 20% Phosal PG in MQ), 100 mg/kg/d navitoclax or venetoclax, 300 or 600 g/kg/d metformin or with drug combinations delivered via intragastric (i.g.) route for 21 days. The cohorts were followed up until 60 days after transplantation. Navitoclax and venetoclax were from AbbVie Inc. Drugs were sonicated daily for better solubility. In the experiments including anti-PD-1, mice received i.p. injections of either 200 μg control IgG (bxcell, #BE0089) or 200 μg anti-PD-1 (bxcell, #BE0146) every third day, in total four times, together with the 1-week daily adjuvant i.g. treatments of either vehicle, venetoclax, metformin or the combination of venetoclax+metformin. Paclitaxel was administrated every third day, i.p., 10 mg/kg in cremphor EL-ethanol-saline. Plasma ALAT levels were measured in the Biochemical Analysis Core for Experimental Research, University of Helsinki.

**Immunoprofiling of mouse peripheral blood samples.** Peripheral blood samples were collected from the tail vein of the mice (21-day drug treatments) into Microtainer blood collection tubes (BD) after which the samples were stained with CD45.1-APC-Cy7 (Biolegend, clone 30-F11, cat. 103116), CD3-APC (Biolegend, clone 17A2, cat.100236), CD4-V500 (BD, clone RM4–5, cat. 560782), CD8b-FITC (eBioscience, clone eBio H35–17.2, cat. 11–0083–85), NK1.1-PE (BD Biosciences, clone PK136 cat. 557391) and red blood cells were lysed after with FACS lysing

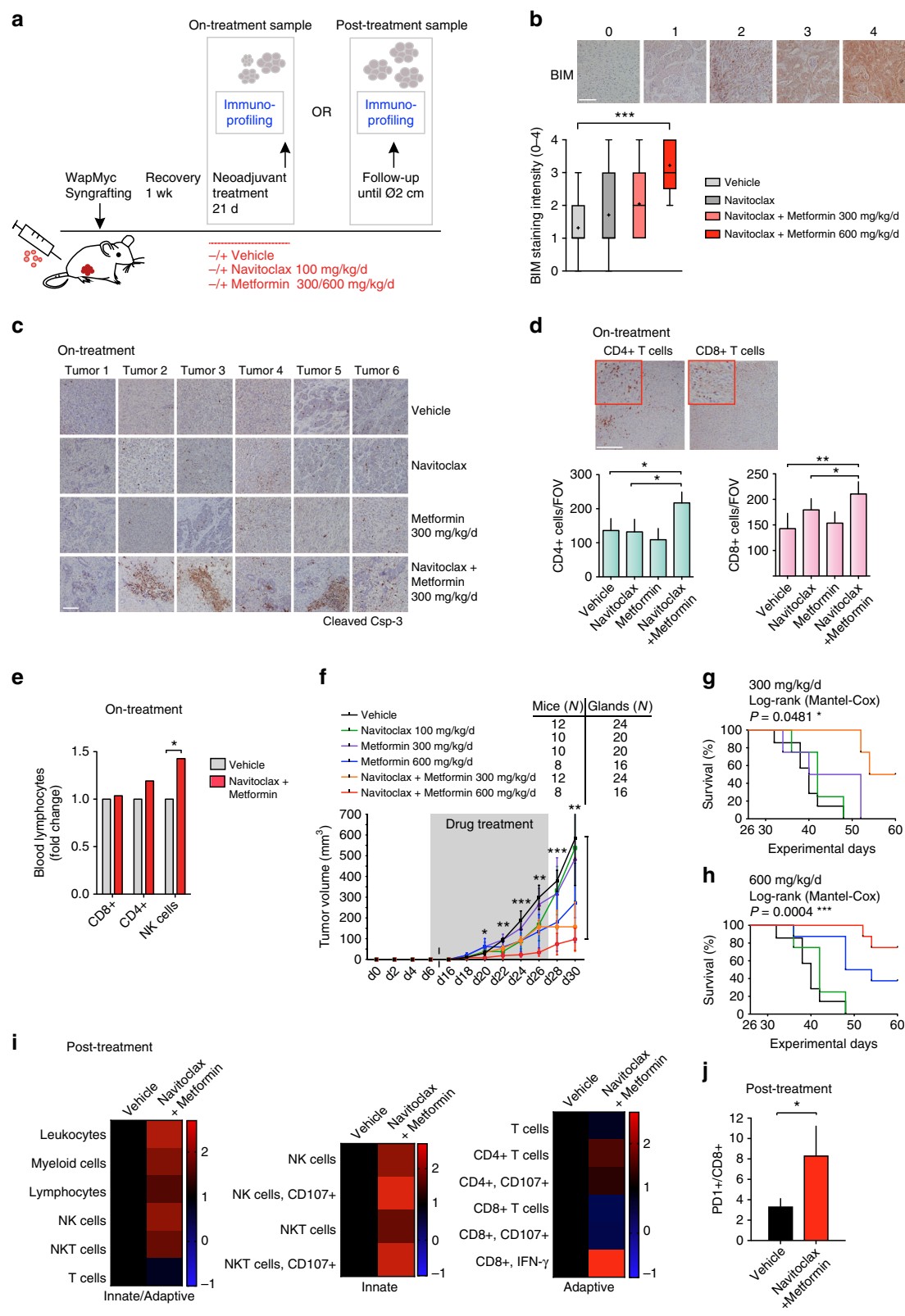

solution (BD). The samples were acquired with FACSVerse (BD Biosciences) and analysed with FlowJo version 10.3 (TreeStar).

**Immunoprofiling of mouse tumor samples**. The tumor homogenate was stained for immunophenotyping by flow cytometry in 3 different tubes each containing

0.067 g of the tumor mass. Following antibodies were used to stain the surface markers: Tube 1: CD45-APC-Cy7, CD3-APC, CD4-V500, CD8b-FITC, PD-1-PerCP-Cy5.5 (BioLegend, clone 29F.1A12, cat.135208), NK1.1-V450 (BD, clone PK136, cat. 560524), CD107a-PE; Tube 2: CD45-APC-Cy7, CD3-APC, NK1.1-PE, CD4-V500, CD8b-FITC, PD-1-PerCP-Cy5.5; Tube 3: CD45-PE (BD, clone 30-F11, cat. 553081), CD11b-BV510 (BD, clone M1/70, cat. 562950), Ly6G-PerCP-Cy5.5

**Fig. 6** ABn treatment inhibits tumor growth and extends survival of mice syngrafted with WapMyc tumors. **a** Combination treatment protocol for orthotopically syngrafted WapMyc tumors. The mice were treated with a vehicle, navitoclax, and two concentrations of metformin alone or in combination with navitoclax for 21 days. Immunoprofiling samples were collected right after the drug treatments or after a follow-up period. In the follow-up period the samples were collected when the tumors reached Ø 2 cm and the mice had to be killed. **b** ABn treatment-induces BIM activation in vivo. The level of BIM immunostaining was scored in IHC samples from treated tumors using a scale of 1–4 (representative examples are shown in the figure). The samples were blinded for analysis. Student's $t$-test (unpaired), SD. **c** ABn treatment-induced "apoptotic ponds". Representative images of tumor samples stained for cleaved caspase-3. **d** ABn treatment stimulates T-cell infiltration. WapMyc tumor samples were isolated from mice killed after the 21-day AB treatment period. Representative images are shown and the data below show average and SD of CD4+ or CD8+ T cells in the tumors. Immunohistochemical stainings were performed for at least 45 tumors per antibody and 3–6 field of view (fov) per tumor were analyzed. Student's $t$-test (unpaired). **e** ABn treatment-induced changes in proportions of peripheral lymphocytes. N (vehicle) = 7 mice, N (ABn) = 4 mice. Blood was collected after 21 days treatment with either vehicle or ABn. Student's $t$-test (unpaired), SD. **f** Tumor growth in ABn-treated mice. Student's $t$-test (unpaired), SEM. **g, h** ABn treatment extends survival. P-values: Difference between the vehicle and ABn-treated cohorts. **i** Flow cytometry-based immunoprofiling of vehicle and ABn (navitoclax+metformin)-treated tumors. The heatmaps show fold-change compared to control. N = 6 (vehicle), N = 2 Navitoclax+Metformin. **j** Post-treatment ratios of tumor-infiltrating PD-1+CD8+ T cells. Student's $t$-test (unpaired), SD

(BD, clone 1A8, cat. 560602), Ly6C-AlexaFluor700 (BD, clone AL-21, cat. 561237), PD-L1-BV421 (BD, clone MIH5, cat. 564716), CD86-APC (BD, clone GL1, cat. 558703) and CD80-PeCy7 (BioLegend, clone 16–10A1, cat. 104734). For intracellular staining the cells from tube 2 were fixed and permeabilized with Cytofix/Cytoperm (BD) according to the kit's instructions and stained with IFN-γ-BV421 (BD, clone XMG1.2, cat. 563376) and Granzyme B-PeCy7 (eBiosciences, clone NGZB, cat. 25–8898–82). Countbright beads (Life technologies) were added to the sample tubes and the cell counts were normalized to 2000 beads counted during the sample acquisition performed with FACSVerse BD. The data were analysed with FlowJo version 10.3. Results are shown using the absolute number of cells, generally excluding cells belonging to parental populations with less than 90 cells.

**Patient-derived explants in 3D culture**. Patient tumor samples were collected from consenting patients with permission, approved by the Hospital District of Helsinki and Uusimaa (Ethical permit: 243/13/03/02/2013). A small sample was cut out from a primary tumor obtained from breast cancer surgery and transferred to the laboratory, typically in less than 30 min. The sample was divided into three parts: One part was snap frozen for DNA/protein analyses, one part was embedded in paraffin for immunohistochemical analyses, and one part was processed for the three-dimensional (3D) culture. The 3D culture sample was carefully minced with a blade and incubated O/N with gentle shaking (130 rpm)+37 °C in Mammocult basal medium (StemCell Technologies) containing 0.2% of Collagenase A (Sigma), Mammocult proliferation supplements (StemCell Technologies), 4 µg/ml heparin, 0.48 µg/ml hydrocortisone, 50 µg/mL gentamicin, and penicillin/streptomycin (all from Sigma). On the following day the mixture was centrifuged at 1400 rpm for 5 min and the pellet was resuspended in 5 ml PBS. Fragments were recentrifuged, resuspended in Matrigel (BD) and seeded to 8-well chamber slides (Nunc) for 3D culture. Samples were cultured in the Mammocult medium described above.

**Breast cancer tissue microarrays**. Breast cancer tissues of 285 women were collected from the archives of the Department of Pathology, Helsinki University Central Hospital. Tissue microarrays were constructed and molecular subtype determined as described in detail earlier[58]. For the analyses, patients whose age at the time of diagnosis or survival was not known, who had lobular or ductal carcinoma in situ, or whose immunohistochemical results for MYC staining were not available or tissue array spot not representative of the tumor were excluded. Altogether, 231 samples were available for analysis, with three replicate samples per patient. BCL-2, BCL-X$_L$, and MCL-1 were scored using reference images as a guide to help the scoring. Human breast cancer TMA samples were obtained and used under ethical permission (HUS 354/13/02/08).

**Cell culture**. MCF10A cells were obtained from ATCC and HEK293 cells were provided by Prof. Timo Otonkoski (University of Helsinki). MCF10A MycER cells (MycER being a tamoxifen inducible MYC fusion protein) have been previously described[14]. MCF10A cells were cultured in epithelial cell basal medium MCDB 170 (US Biological) supplemented with 5 µg/mL insulin, 70 µg/mL Bovine Pituitary Extract (BPE, Sigma), 0.5 µg/mL hydrocortison, 5 ng/mL epithelial growth factor (EGF), 5 µg/mL human transferrin, 0.01 µM isoproterenol, and antibiotics. HEK293 cells were cultured in DMEM with 10% FBS, ʟ-glutamine and penicillin/streptomycin.

**TNBC cell line panel**. A panel of 17 TNBC cell lines was obtained from ATCC (ATCC-TCP-1003). Cell lines and culture medium compositions are presented in Supplementary Table 2.

**Induction of MYC with CRISPR/dead-Cas9**. PiggyBac construct PB-tetON-DDdCAS9VP192-T2A-GFP-IRES-Neo[23] was modified to incorporate 2 additional transactivation domains, p65 and HSF1, PCR-cloned from lenti sgRNA(MS2)_zeo

plasmid (Addgene, #61427,[59]). We designed five guide RNAs targeting the MYC promoter, prepared gRNA-PCR transcriptional units and concatenated using Golden Gate assembly. The concatenated MYC guide RNAs plasmid was cloned into a PiggyBac backbone to obtain PB-GG-CMYC-PGK-Puro transposable plasmid. For the generation of a stable cell line with an inducible MYC dCas9-activation system, cells were transfected or transduced with PB-tetON-DDdCAS9VP192-p65-HSF1-T2A-GFP-IRES-Neo, PB-GG-MYC, and PB-CAG-rtTA together with a PiggyBac-transposase plasmid, using FuGene HD (Promega). After 72 h the cells were selected with G418 and puromycin. An inducible lentiviral version of the dCas9-activator (FU_tetO_dCas9VP192-T2A-EGFP) was generated by cloning dCas9VP192-T2A-EGFP fragment into an FUW-tetO lentiviral backbone (Addgene, #41084). We generated a lentiviral vector for the delivery of sgMYC1 (LentigRNA_CMYC_1) by cloning MYC_1 transcriptional unit PCR fragment into LentiCRISPR v2 backbone (Addgene, #52961). The sgRNA sequences are shown in Supplementary Table 4.

**Cell viability assays**. TNBC cell lines were plated on poly-D-lysine coated 96-well plates (354640, Corning BioCoat), 150 µl/well, and incubated at 37 °C for 24 h 50 µl of 0.5% DMSO control, positive controls 0.05 ng/µl Nocodazole (S2775, Seleckchem), and 10 nM paclitaxel (S1150, Selleckchem), and navitoclax, metformin, A-769662 (MedChemtronica) test concentrations were added to the cells in four replicates, and the cells were treated for 72 h. For combination treatments navitoclax EC$_{20}$ (unique for each cell line) concentration was tested in combination with metformin and A-769662 test concentrations. Cell viability was determined by adding 20 µl/well alamarBlue reagent (88952, Thermo Scientific). Fluorescence was measured using a FLUORstar Omega plate reader, with 560 EX nm/590 EM filter settings, 4 h, 6 h, and 8 h after alamarBlue addition.

Caspase-Glo® 3/7 Assay (Promega) was performed according to manufacturer's instructions and luminescence (RLU) was measured using VICTOR™x3 plate reader (Perkin Elmer).

**P53-long range PCR and sequencing from patient samples**. KAPA Long Range HotStart DNA Polymerase (KAPA Biosystems) was used to amplify the Tp53 gene from the genomic DNA. Three PCR primer pairs were designed to cover the whole Tp53 sequence. PCR products were purified with the Agencourt AMPure XP PCR Purification systems (Beckman Coulter) and quantified using the Qubit dsDNA BR Assay system (Invitrogen). The sequencing libraries were constructed with a Nextera XT library preparation kit (Illumina) and the pooled, barcoded libraries were subsequently sequenced using the MiSeq sequencer with 300PE reads. The basic processing consisted of data quality analysis (FastQC), trimming of the poor quality and adapter sequences (Trimmomatic), alignment of the sequences against the hg19 reference genome using BWA (Burrows-Wheeler Aligner) and alignment map sorting and duplicate identification by Picard tools (SAMTools). The actual analysis consisted of GATK (the Genome Analysis Toolkit) with local realignment around indels, base quality score recalibration, and variant calling, according to the GATK Best Practices recommendations. Illumina VariantStudio was used to annotate the SNPs and indels discovered. Sequencing primers are listed in Supplementary Table 4.

**Patient-derived xenograft sequencing**. Total RNA was extracted from snap frozen samples using the mirVana miRNA Isolation kit (Ambion, Carlsbad, CA) according to the manufacturer's instructions. Genomic DNA was removed using the RNase-free DNaseSet (Qiagen). The quality of the RNA preparations was controlled using a Bioanalyzer (Agilent Technologies). For DNA isolation, snap-frozen tumor samples were digested with proteinase K at 55 °C overnight and lysates were digested with RNase A (Qiagen). Next, the DNAs were extracted using phenol:chloroform:isoamylalcohol and precipitated with ethanol. The DNA pellets were washed and resuspended in TElow buffer (Tris 10 mM, pH 8, EDTA 0.1 mM, pH 8). The integrity of the DNA samples was checked after 1.3% agarose gel

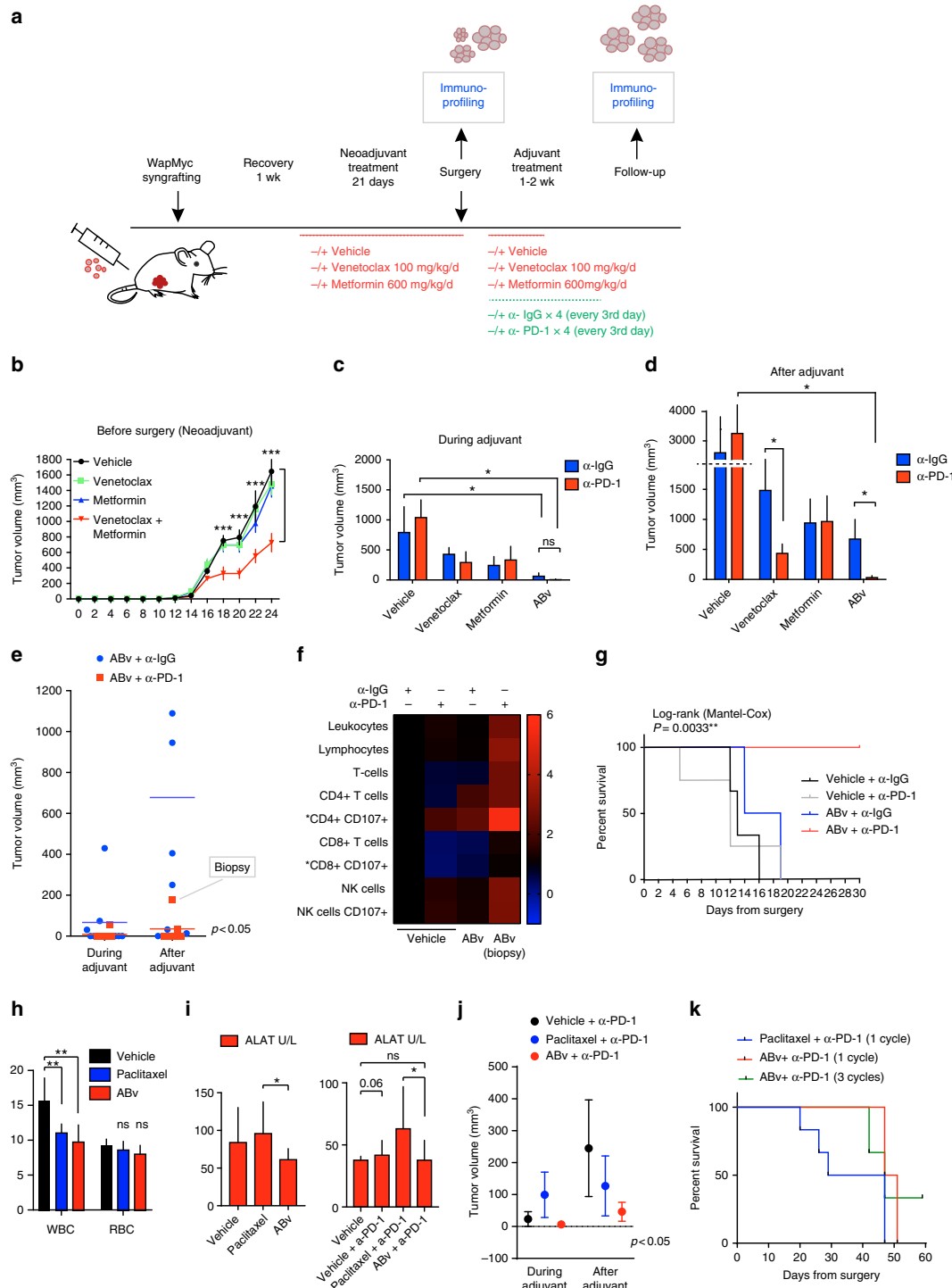

electrophoresis, and the purity of the DNA samples was determined using a NanoDrop 2000 system (Thermo Scientific, Waltham, MA). All PDX models were analyzed by Affymetrix HGU133 plus2.0, Affymetrix SNP6.0 and using whole exome sequencing with Illumina HiSeq2000 at GATC (Konstanz, Germany).

**Immunohistochemistry**. Tissues were collected and fixed with 4% paraformaldehyde (PFA) and embedded in paraffin. In all, 3–5 μm sections were cut and the slides were deparafinized and incubated in 1X Antigen retrieval citrate buffer solution (Dako) in +70 °C overnight. Histochemical stainings were carried out using standard techniques. Images were taken with a Leica DM LB microscope and Studio-Lite 1.0 software (Biomedicum Imaging Unit, University of Helsinki). Images and slides were blinded for analysis.

**Immunofluorescence stainings**. Cells grown and treated on coverslips were fixed with 4% PFA and permeabilized with 0.1% Triton X-100. 2D immunofluorescence stainings were performed using standard protocols. 3D cultured patient samples were fixed with 2% PFA, permeabilized with 0.25% Triton X-100, and blocked with 10% goat serum in PBS. Primary antibodies were diluted in blocking buffer and 3D cultures were stained overnight. After three washes with IF buffer (0.1% BSA, 0.2% Triton X-100, 0.05% Tween-20 in PBS), the structures were incubated with secondary antibodies in blocking buffer and washed again. Nuclei were counterstained with Hoechst 33258 (Life Technologies) and mounted with Immu-Mount reagent (Fisher Scientific). Imaging was performed using a Leica TCS CARS SP8 confocal microscope (Biomedicum Imaging Unit, University of Helsinki).

**Cell lysis and western blot**. Isolated WapMyc tumor cells were lysed with non-ionic Triton X-100 cell lysis buffer (10 mM Tris, pH 7.5, 130 mM NaCl, 1%

**Fig. 7** ABv treatment with anti-PD-1 immunotherapy offers durable therapeutic response. **a** Treatment protocol for syngrafted WapMyc tumors. The mice received vehicle, venetoclax, metformin, or venetoclax+metformin (ABv) for 21 days. Tumors were surgically removed, and mice were given adjuvant treatments. ABv was given every day for 1 week, and control IgG or anti-PD-1 was given every third day, four times in total. The mice were followed up until the tumors reached ∅ 2 cm. Flow cytometry-based immunoprofilings were done after the adjuvant treatment and after the follow-up. **b** Tumor growth during the neoadjuvant treatment. Student's t-test (unpaired), SEM. N = 8 mice/treatment group. **c** Tumor volume during the adjuvant treatment (d7 from surgery). Student's t-test (unpaired), SD. N = 4 mice/treatment group. **d** Tumor volume after the adjuvant treatment (d14 from surgery). Student's t-test (unpaired), SD. **e** Box plot of tumor volumes in ABv+IgG- vs anti-PD-1-treated groups during and after the adjuvant treatment. After the adjuvant treatment, the only tumor in the ABv+anti-PD-1 group was surgically biopsied for immunoprofiling. **f** Tumor immunoprofiles after the adjuvant treatment. The heatmap shows fold-changes compared to vehicle+IgG. N = 4 mice/group, except N = 1 in the biopsy. Cell populations derived from parental populations with less than 90 cells are marked with asterisk (*). **g** Survival of the mice. P-value: Difference between vehicle+IgG and ABv+anti-PD-1-treated mice. **h** Total white blood cell (WBC) and red blood cell (RBC) counts in the treatment groups. Student's t-test (unpaired), SD. **i** Liver injury measured by plasma ALAT levels. Student's t-test (unpaired), SD. **j** Average tumor sizes during (d11) adjuvant treatment with vehicle+anti-PD-1, paclitaxel+anti-PD-1, or ABv+anti-PD-1, and after the drug withdrawal (d15). Length of one treatment cycle is 10 days. p-value denotes the difference between post-adjuvant vehicle+anti-PD-1 and ABv+anti-PD-1. N = 3 (vehicle+anti-PD-1), 7 (Paclitaxel+anti-PD-1), 10 (1 cycle of ABv+anti-PD-1). SD, student's t-test. **k** Survival of the mice. N = 6 (Paclitaxel+anti-PD-1), 2 (1 cycle of ABv+anti-PD-1), 3 (3 cycles of ABv+anti-PD-1)

---

Triton X-100, 10 mM NaPi, 10 mM Nappi). The lysed cells were incubated on ice for 10 min and centrifuged at 13,400 rpm for 15 min at +4 °C. Snap-frozen tissue samples (3 × 3 mm) were placed in homogenization tubes containing ceramic beads (Bertin Technologies) and 300 μl of mammary gland lysis buffer (10 mM Tris-HCl, pH 7.4 (HusLab), 150 mM NaCl (Sigma), 1% Triton X-100 (Sigma), 1 mM EDTA (Sigma), 1 mM EGTA (Sigma), 0.5% NP-40 (Fluka), and 1 protease inhibitor coctail tablet (Roche) per 10 ml of solution) and homogenized with Precelly's homogenizer (Bertin Technologies). After homogenization, the sample tubes were kept on ice for 30 min and then centrifuged 16,000 rpm for 30 min at +4 °C. The cultured cell lines were lysed with RIPA lysis buffer. SDS-page and western blot were done using standard protocols. Main manuscript western blots with original ladders are shown in Supplementary Figure 6.

**Annexin V/PI assay.** WapMyc and normal porous cells were isolated from the animals and cultured in low adhesion for 24 h and treated with drug treatments. After treatments the samples were treated briefly with StemPro Accutase (Gibco) to detach the cells from mammospheres. Cells and medium were collected and Annexin V-FITC/PI assay was performed according to manufacturer's instructions (BioVision). Flow cytometry was performed using BD Accuri C6 flow cytometer in Biomedicum FACS Core.

**Antibodies.** Antibodies used for western blot, immunohistochemistry, and immunofluoresecence are shown in Supplementary Table 3.

**Data analysis.** GraphPad Prism 5 software was used for statistical analyses.

**Reporting summary.** Further information on experimental design is available in the Nature Research Reporting Summary linked to this article.

## Data availability
Sequencing data have been deposited to the Sequence Read Archive (SRA) with accession number: SRP178687 (BioProject ID: PRJNA514437). Relevant manuscript data are available from the authors upon request.

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

## Acknowledgements

We thank Biomedicum Functional Genomics Unit (FuGu), Biomedicum Imaging Unit (BIU), University of Helsinki Language services, and Biomedicum FACS Core for services, Tarja Välimäki and Maiju Merisalo-Soikkeli for technical support, Prof. Kari Alitalo and Pauliina Kallio (University of Helsinki) for providing advice and reagents for mouse experiments, Anne Lehtonen (AbbVie Oy) for commenting the manuscript, and Johanna Englund for help with the experiments. This work was funded by the Academy of Finland, TEKES and the Finnish Cancer Organizations; H.M.H. was funded by the Integrative Life Sciences Doctoral Program (ILS), DSHealth doctoral school, Biomedicum Helsinki Foundation, Orion-Farmos Foundation, Emil Aaltonen Foundation, and the Inkeri & Mauri Vänskä Foundation. S.M. and M.I. were funded by the Finnish Cancer Institute, Sigrid Juselius Foundation, Finnish special governmental subsidy for health sciences, research, and training (EVO-funding).

## Author contributions

H.M.H., J.K and J.M.A. designed the study, provided and analyzed the data, and wrote the manuscript. E.M., D.B., B.v.E., J.D.L., S.M., P.K and M.E. edited the manuscript, and provided and analyzed the data. H.M.H., J.M.A., E.M., T.R., M.I., H.H., H.A.H., M.S., V.E., P.M., A.N. and J.S. performed experiments and analyzed the data. T.D.L., T.A., and H.S. analyzed the data. T.O., P.H., M.L., J.M. and H.J. provided reagents, helped with analysis, clinical data, and patient samples. P.H., M.L., H.J., S.M., P.K., M.E. and J.D.L. equally contributed as senior authors of the manuscript.

## Additional information

**Competing interests:** The in vivo work was partly funded by AbbVie Inc. J.D.L. is an employee and a shareholder in AbbVie Inc. The remaining authors declare no competing interests.

