## [Peer Review File · Nature Communications]

Reviewers' comments:

Reviewer #1 (Remarks to the Author):

I enjoyed reading the manuscript by Haikala and colleagues. It represents a substantial amount of work which demonstrates that co-targeting of AMPK and BCL2 pathways. The experiments and approaches are of high quality and the effects that they observe with combination of AB+PD1 therapy (Figure 7), if replicated in human patients could move clinical therapies forward significantly. I am thus generally in favor of publication, though I think some issues should be addressed:

Major Critiques:

- the set of experiments that really raise excitement about the studies are shown in Figure 7H where relapse is noted to be markedly diminished or perhaps block almost entirely. However, it is difficult to judge how truly revolutionary the treatment regimen is since the cohort were only followed for 30 days. Similar treatment should be carried out for much longer to determine if mice treated in this manner with combinatorial treatments are indeed 'cured'.
- along these lines, most of the in vivo studies are intended to show a diminished rate of tumor formation, i.e. a prevention type of trial, where cell or tumors are transplanted and then very soon thereafter treated. It is therefore not clear if established tumors would respond to this treatment strategy.
- MYC dependence: the authors do a nice job to show that ABn treatment preferentially kills MYC ON cells versus those with lower MYC expression. However, the in vivo studies do not explore this concept. If the authors seek to make the claim that the combination of AB+antiPD1 is MYC-selective or MYC-dependent then they would need to do additional studies. For PDX studies, only one line which shows some level of MYC staining is shown. What about a PDX with lower MYC expression, would this similarly respond to AB therapy, and if it does, what would the authors conclude? Similarly, for the AB+antiPD1 treatments only one transgenic model is used which is driven by MYC. What about MMTV-PyMT models that don't rely on MYC as the principal driver, would this similarly respond to this treatment strategy?
- based on the authors rationale for immunotherapy experiments, arguably any drug that induces a bit of apoptosis in the primary tumor should sensitize to PD1 antagonists. If so then using a standard of care anthracycline or taxane would be helpful to compare to the AB+PD1 treatment strategy. Is there something unique about this approach or will any treatment that partially kills the tumor cells give a similar response?
- the authors also identify several other combinations that preferentially kill MYC high MCF10A cells + ABT737 (i. Nutlin, Prima, MK2206, etc). While the AMPK activator is marginally more active in this one assay, what do the authors think about these other treatment approaches? This should be discussed in more detail. As written it seems that nearly anything added to the ABT compound gives additional activity.

Reviewer #2 (Remarks to the Author):

This is a carefully documented study on the relationship of MYC to susceptibility to Anti-PD1 immunotherapy in breast cancer. The results are of potential interests. However, the writing is very dense, confusing and hard to read, and detailed. It is repetitive since much of the data information in the text is also in the figures. Many experiments are included that might be better put in the supplement. It would be best of the results consistently use one inhibitor rather than changing them around. Extensive editing and shortening would improve this manuscript substantially.

Specific comments:

1. Fig. 1 and associated text. The difference in tumor burden without any effect on survival is strange.
2. On page 7 and Fig. 2 the text states "The type II diabetes drug metformin, another AMPK activator, also induced MYC-dependent apoptosis when combined with ABT-737 (Fig.2G). Even high concentrations of A-769662 did not induce detectable cell death with MYC, indicating synthetic lethal interaction of AMPK activation and ABT-737 with MYC". This is confusing since it looks like both Metformin and A-769662 have the same effects (Fig. 2F and G).
3. MCF10A cells are used in experiments described in fig 2 and 3. Where these experiments also done in tumor cells that are Myc+ or -. Although the Fig. 2 legend says HEC293 cells were also used, no data for these cells are shown in the figure.
4. The data in Fig. 5H, 6G and H and Fig. 7G are inconsistent. Why are there different efficacies
5. Why are different inhibitors used in the experiments in fig. 7 (Venetoclax) and Fig. 6 (Navitoclax) ?

Minor comments

There are a number of incomplete sentences, tense and other editing errors: e.g., paragraph 1 in the Results section, the last sentence does not make sense "Thus, high expression levels of MYC and anti-apoptotic BCL-2 proteins is a common potentially features in breast cancer."

Reviewers' comments:

Reviewer #1 (Remarks to the Author):

I enjoyed reading the manuscript by Haikala and colleagues. It represents a substantial amount of work which demonstrates that co-targeting of AMPK and BCL2 pathways. The experiments and approaches are of high quality and the effects that they observe with combination of AB+PD1 therapy (Figure 7), if replicated in human patients could move clinical therapies forward significantly. I am thus generally in favor of publication, though I think some issues should be addressed:

Major Critiques:

1. The set of experiments that really raise excitement about the studies are shown in Figure 7H where relapse is noted to be markedly diminished or perhaps block almost entirely. However, it is difficult to judge how truly revolutionary the treatment regimen is since the cohort were only followed for 30 days. Similar treatment should be carried out for much longer to determine if mice treated in this manner with combinatorial treatments are indeed 'cured'.

Author's response:

We thank the reviewer for suggesting this highly relevant experiment. For revision, we repeated the experiment shown in Fig. 7G with longer follow-up period. We also tested the effects of a single cycle ABv+anti-PD1 treatment versus 3 cycles of adjuvant ABv+anti-PD1 treatment. The new experiment shows that continued ABv+anti-PD1 treatment improves the outcome but also that the mice were not cured from transplanted tumors. In the end of the follow-up period, we had one survivor mouse left and this one had received triple cycle of ABv+anti-PD1 as a treatment.

We think that the mouse experiments will not yield much further information or predictions about the potential efficacy of the treatment in humans, so the next steps must involve explorative clinical trial to look for positive signals in patients.

We show the new data in Fig 7H-K added to the manuscript.

NEW TEXT:

Results:

Page 15: The ABv+anti-PD-1 treatment offered striking survival benefit, with all the mice in the group surviving through the 30-day follow-up period (**Fig.7G**).

Next, we investigated the durability of the ABv + anti-PD-1 treatment response more thoroughly, extending the follow-up period to 60 days, and also assessed its efficacy in comparison to the combination of paclitaxel and anti-PD-1 antibody. Taxanes (including paclitaxel) are standard of care in the treatment of patients with breast cancer and encouraging results were reported recently from the Phase III IMpassion130 study, which included combination of nab-paclitaxel plus the anti-PD-L1 antibody atezolizumab for patients with advanced TNBC (Leisha., 2016). We noted that both ABv and paclitaxel treatment caused low white blood cell counts (**Fig.7H**); however, only the paclitaxel+anti-PD1 combination caused significant elevation of alanine transaminase (ALAT), a marker of liver injury (**Fig.7I**). Tumor growth was clearly restricted in cohorts that received either ABv+anti-PD1 or paclitaxel+anti-PD1 (**Fig.7J**). Comparing the treatments, the therapeutic benefit was similar in both tumor growth and survival analyses (**Fig.7J-K**). One cohort of ABv+anti-PD1 treated mice was divided into two smaller cohorts - one receiving three cycles of ABv+anti-PD1 and the other one only one cycle as in the previous experiment. Only one mouse survived to the end of the 60-day follow-up period, and this one had received three cycles of ABv+anti-PD1 (**Fig.7K**). Although ABv+anti-PD1 treatment does not cure mice syngrafted with aggressive tumors, these data indicate that it significantly delays tumor re-growth and markedly improves their survival.

In conclusion, we propose that the combination of cell death-inducing AB therapy together with anti-PD-1 immunotherapy provides a potent new tumor-targeting strategy with high translational potential in tumor types with high MYC expression.

Figure 7H-K:

H) Total white blood cell (WBC) and red blood cell (RBC) counts in the treatment groups. Student's *t*-test (unpaired), SD. (I) Liver injury measured by plasma ALAT levels. Student's *t*-test (unpaired), SD. (J) Average tumor sizes during (d11) adjuvant treatment with vehicle + anti-PD1, paclitaxel + anti-PD1, or ABv + anti-PD1, and after the drug withdrawal (d15). Length of one treatment cycle is 10 days. *p* value denotes the difference between post-adjuvant vehicle+anti-PD1 and ABv+anti-PD1. N=3 (vehicle + anti-PD1), 7 (Paclitaxel + anti-PD1), 10 (1 cycle of ABv+anti-PD1). SD, student's *t* test. (K) Survival of the mice. N=6 (Paclitaxel + anti-PD1), 2 (1 cycle of ABv+anti-PD1), 3 (3 cycles of ABv+anti-PD1). Animals which formed tumors in the surgery scar were excluded from the experiment.

Materials and methods:

Page 39: Paclitaxel was administrated every third day, i.p., 10 mg/kg in cremphor EL-ethanol- saline. Plasma ALAT levels were measured in the Biochemical Analysis Core for Experimental Research, University of Helsinki.

2. Along these lines, most of the in vivo studies are intended to show a diminished rate of tumor formation, i.e. a prevention type of trial, where cell or tumors are transplanted and then very soon thereafter treated. It is therefore not clear if established tumors would respond to this treatment strategy.

Author's response:

The reviewer points out a well-recognized problem with mouse tumor models and undoubtedly, the bias has led to many overinterpretations with regards to anti-cancer potential of new drugs. However, we believe that our studies in syngrafted WapMyc syngraft model are not surrogates of prevention but rather they estimate the ability of a given drug to control tumor growth.

We graft about 200,000 cells per gland, which preserves great deal of the original tumor heterogeneity. Typically, with these numbers, the mice have visible or palpable tumors at the start of the treatment. We have estimated that the detection limit of palpation is about 1x1 mm = 1 mm³ tumor size and we usually detect this or bigger size tumors at the start of the treatment (see Figure below). In repeated experiments, 100% of the mice develop tumors after grafting.

While these notions argue that we have probed ABv action in established tumors, it is clear that the models are just models, which will never be able to fully predict drug efficacy in clinics.

Figure for the reviewer only: Mice grafted with WapMyc tumor cells at the start of the treatment (7 days from tumor cell grafting).

3. MYC dependence: the authors do a nice job to show that ABn treatment preferentially kills MYC ON cells versus those with lower MYC expression. However, the *in vivo* studies do not explore this concept. If the authors seek to make the claim that the combination of AB+antiPD1 is MYC-selective or MYC-dependent then they would need to do additional studies. For PDX studies, only one line which shows some level of MYC staining is shown. What about a PDX with lower MYC expression, would this similarly respond to AB therapy, and if it does, what would the authors conclude? Similarly, for the AB+antiPD1 treatments only one transgenic model is used which is driven by MYC. What about MMTV-PyMT models that don't rely on MYC as the principal driver, would this similarly respond to this treatment strategy?

Author's response:

The establishment of MYC-dependence *in vivo* (or *in vitro*) is obviously a challenging task since MYC inhibition is rather expected sensitize to than block apoptosis (due to addiction). However, it is a fair point that additional studies with "MYC low" models could shed light to the question of MYC-selectivity. We focused the analysis on primary

transplanted tumor growth since tumor removal turned out to be very difficult in the models we analyzed below.

We identified one PDX model with "low" MYC expression and also recruited FVB MMTV-PyMT tumor model to our assays. Both of these new models showed scattered, infrequent MYC expression in IHC, which pattern was clearly different from earlier models judged as "MYC high" (revision **Fig. S5F-G**). The tumor cells from these mice were orthotopically transplanted into the mouse mammary glands of FVB mice (MMTV-PyMT) or NSG mice (pdx) and treated as before. The results are shown in new **Fig. S5F-G**. In contrast to WapMYC driven tumors, the MMTV-PyMT tumors did not respond to navitoclax or metformin as single agents or in combination. In the MYC-low PDX model, the tumors responded well to navitoclax as a single agent, but there was no additional benefit from combination with metformin. These results are consistent with the notion that MYC high expression status is critical in the present treatment strategy.

NEW TEXT:

Page 13: ABn treatment does not inhibit tumor growth in breast cancer models with low Myc expression status

To address the importance of high MYC expression for the efficacy of the ABn regimen, we employed two mouse models of breast cancer with low MYC expression, the MMTV-PyMT model with tumors syngrafted to FVB mice and an ER+ PDX model with tumors grafted to NSG mice. Mice with orthotopically engrafted tumors were subjected to the ABn combination as before. In contrast to FVB WapMYC syngrafts, the FVB MMTV-PyMT tumors did not respond to navitoclax or metformin, either as single agents or in combination (**Fig S5E**). Although tumors in the MYC-low ER+ PDX model responded well to navitoclax as a single agent (we note that BCL-2 expression is often reported as enriched in this breast cancer subtype; Deng and Letai, 2013), combining it with metformin did not offer any added benefit (**Fig S5F**). These results are consistent with the notion that high MYC expression defines tumors that are most likely to respond to ABn treatment.

New Figure S5F-G (added in Figure S5): (F) Efficacy of navitoclax + metformin treatment in MMTV-PyMT syngraft model with low MYC expression status. Immunohistochemical staining of MYC shown in the right. (G) Efficacy of navitoclax + metformin treatment in MYC-low ER+ PDX model. Immunohistochemical staining of MYC shown in the right.

Figure for reviewer to demonstrate IHC landscape in MYC-high tumors (Figure 1).

Materials and Methods:

For ER+ PDX, 60-day 17 β estradiol pellets (Innovative Research America) were inserted into the neck of the animals to promote tumor growth. The orthotopic syngrafting of MMTV-PyMT tumor cells to FVB hosts was performed similarly as described for FVB WapMyc syngrafts. The tumor cells were isolated from MMTV-PyMMT donor and 1.000.000 cells were transplanted per gland.

4. Based on the authors rationale for immunotherapy experiments, arguably any drug that induces a bit of apoptosis in the primary tumor should sensitize to PD1 antagonists. If so then using a standard of care anthracycline or taxane would be helpful to compare to the AB+PD1 treatment strategy. Is there something unique about this approach or will any treatment that partially kills the tumor cells give a similar response?

To address this highly relevant question, we performed the experiments that have been described in our response to point 1. The benefits of the ABV+anti-PD1 treatment over paclitaxel+anti-PD1 in mouse experiments include less liver injury and better efficacy. While we are truly excited about these results, it is clear that mouse experiments may not fully predict situation in patients. Therefore, further validation of the potential treatment efficacy and low toxicity needs to be performed at the level of explorative clinical trial.

5. The authors also identify several other combinations that preferentially kill MYC high MCF10A cells + ABT737 (i. Nutlin, Prima, MK2206, etc). While the AMPK activator is marginally more active in this one assay, what do the authors think about these other treatment approaches? This should be discussed in more detail. As written it seems that nearly anything added to the ABT compound gives additional activity.

We realize that we may not have communicated well the rationale of the screen in the text. It is well established that ABT-737 synergizes with wide variety of drugs in induction of cell death and we were not interested in combinations with maximal apoptotic activity. Instead, we were looking for maximal difference between -MYC and +MYC conditions. In other words, maximal synthetic lethal activity of two compounds with MYC.

In this context, the activities of Nutlin-3a, PRIMA-1 and A-769662 (combined with ABT-737), were most interesting.

To communicate better the rationale of the screen and to discuss of the other hits too (p53 drugs), we have made the following changes in the text:

NEW TEXT:

Page 7: The apoptotic potential of single-agent and combination treatments was tested with and without MYC activation (**Fig.2B-C**). All tested compounds potentiated the apoptotic action of ABT-737 but the level of MYC-dependence varied. Combinations with BEZ-235 and MK2206 were active regardless of MYC status (**Fig.2C**) and were not investigated further. Interestingly, a strict MYC-dependency was observed for apoptosis induced by Nutlin-3a and PRIMA-1, which both act by inducing p53 activity, and A-769662, a compound that activates AMPK by an allosteric mechanism (**Fig.2D**). These observations are consistent with earlier findings showing that the MYC-dependent pro-apoptotic action of AMPK is coupled with p53 in MCF10A cells (Nieminen). A-769662 did not induce cell death without sensitization by MYC and ABT-737, even at high concentrations (**Fig.2E-F**). Metformin, a commonly used type II diabetes drug that also activates AMPK, induced similar MYC-dependent apoptosis when combined with ABT-737 (**Fig.2G-H**).

The order of the figures have been changed in the revision.

Reviewer #2 (Remarks to the Author):

This is a carefully documented study on the relationship of MYC to susceptibility to Anti-PD1 immunotherapy in breast cancer. The results are of potential interests. However, the writing is very dense, confusing and hard to read, and detailed. It is repetitive since much of the data information in the text is also in the figures. Many experiments are included that might be better put in the supplement. It would be best of the results consistently use one inhibitor rather than changing them around. Extensive editing and shortening would improve this manuscript substantially.

Specific comments:

1. Fig. 1 and associated text. The difference in tumor burden without any effect on survival is strange.

The most accurate part of the tumor measurement spans the time before any of the mice had to be sacrificed due to tumor burden. In Fig. 1G, the most reliable part spans first 12 days of the treatment and in Fig. 1K, the first 17 days.

After mice started to drop off from the cohorts, the tumor growth measurements were not able to represent reliably the average tumor growth in the population. Therefore, we have greyed the unreliable part of the curve (explained in the text).

To rationalize the somewhat discordant sounding results from tumor growth assays (small but statistically significant difference) and survival assays (no statistically significant difference, although small trend can be seen in Fig 1K), we think that **in short-term** there is a small difference in the speed of tumor growth but **in long-term** the tumors reach the maximum size in more or less same time. There was a clear signal from ABT-737 in the original tumor growth assays but it is quite evident from all

assays that the effect was not strong enough to provide any significant long-term survival benefits.

We have edited the text to clarify the difference between tumor growth versus survival assays:

NEW TEXT:

Page 6: ABT-737 inhibited tumor growth during the first 12 days of treatment (**Fig.1G**), after which time the first animals had to be sacrificed. Therefore, at later time points the average tumor volume does not indicate tumor growth rates reliably (**Fig.1G** left, grey area). To estimate the tumor growth rates throughout the experimental period, we used a mixed-effects modeling framework that excludes sacrificed animals from the experiment. While the analysis demonstrates that ABT-737 inhibits tumor growth over the whole 21-day treatment period (**Fig.S1E**), there was no statistically significant difference in the overall survival between the control and ABT-737-treated groups (**Fig.1G** right). Interestingly, lung metastases were observed in only 27% of the ABT-737-treated mice, compared to 50% of mice in the control group (**Fig.1H**).

ABT-737 treatment induced apoptosis in these tumors (**Fig.S1F-G**) and slowed tumor growth at early stages of tumor formation (**Fig.1K**, left). Once again there was no difference in the overall survival between the control and the ABT-737-treated groups (**Fig.1K**, right).

In summary, Bcl-2/Bcl-X_L inhibition promotes apoptosis in Myc-driven mouse mammary tumors and inhibits the growth of primary tumor, as well as its dissemination to distant tissues. However, these effects were not strong enough to provide clear survival benefits over time.

2. On page 7 and Fig. 2 the text states “The type II diabetes drug metformin, another AMPK activator, also induced MYC-dependent apoptosis when combined with ABT-737 (Fig.2G). Even high concentrations of A-769662 did not induce detectable cell death with MYC, indicating synthetic lethal interaction of AMPK activation and ABT-737 with MYC”. This is confusing since it looks like both Metformin and A-769662 have the same effects (Fig. 2F and G).

We agree that the wording was confusing. In the revised version, we have changed the text and the order of figures (see our comments to Ref#1 point 5).

3. MCF10A cells are used in experiments described in fig 2 and 3. Where these experiments also done in tumor cells that are Myc+ or -. Although the Fig. 2 legend says HEC293 cells were also used, no data for these cells are shown in the figure.

The experiments in Fig. 2-3 describe the path for discovery of AMPK+Bcl-2/xL synergy in MCF10A cells with high engineered MYC activity. The main discoveries from this part of the study were:

- A-769662+ABT-737 and metformin+ABT-737 promote apoptosis in the cells with high MYC activity
- The apoptotic activity is coupled to induction of BIM

These results were validated in

- PDECs (Fig. 4: Cell death in E-F, BIM response in F; Fig 5. A-B)
- Panel of 16 TNBC cell lines and PDX model of TNBC (Fig 5. cell death). See also new data relating to this figure.
- Wap-MYC syngrafts (Fig. 6 Cell death, BIM; Fig. 7 Cell death together with a-PD-1). See also new data relating to Fig. 6.

Overall, we think that we have very extensively validated our key findings in wide variety of breast tumor models.

The CRISPR/dead-Cas9 system was first established in HEK293 cells since these cells are highly transfectable. However, HEKs are highly transformed. They express SV40 Large T antigen, and hence lack normal pRb and p53 functions and these cells also present many abnormalities in MYC function. Therefore, we tend to use these cells as convenient tools to establish genetic inducible systems but then explore the MYC actions in less transformed systems like MCF10A.

4. The data in Fig. 5H, 6G and H and Fig. 7G are inconsistent. Why are there different efficacies

The experiments have been performed in different tumor models or in different setting. The data between the figures are not cross-comparable.

Fig.5H shows results from ABn treated TNBC PDX model. Fig. 6G shows survival curves corresponding to mice treated with ABn to inhibit primary tumor growth in neoadjuvant setting. Fig. 7G shows survival curves corresponding to mice treated with VeM to inhibit regrowth of tumors (after dissection) in adjuvant setting.

5. Why are different inhibitors used in the experiments in fig. 7 (Venetoclax) and Fig. 6 (Navitoclax)?

The manuscript presents results from a project that has been lasting for 10 years. Initially we used ABT-737 that was available then and later, when the project matured and the need for extensive mouse experiments became apparent, we were sponsored with sufficient quantities of orally bioavailable navitoclax to perform the mouse experiments in a cohort set-up. The prospects of different BCL-2 inhibitors for clinical translation have also changed during the course of these studies. We employed later venetoclax to our studies since this drug has been recently approved for clinical use via several FDA fast-track designations. Venetoclax had weaker but still comparable efficacy to navitoclax in cells and this difference seems to translate to mouse studies. Nevertheless, the apoptotic venetoclax action was sufficient to benefit from anti-PD1. We hope that this clarifies the reasons of using different BCL-2 inhibitors in our study. We think that similar results obtained with slightly different classes of BCL-2 inhibitors strengthens the main hypothesis and demonstrates its robustness.

Minor

comments

There are a number of incomplete sentences, tense and other editing errors: e.g., paragraph 1 in the Results section, the last sentence does not make sense “Thus, high expression levels of MYC and anti-apoptotic BCL-2 proteins is a common potentially features in breast cancer.”

We apologize these mistakes. The manuscript has been now edited and checked for grammatical errors by the University of Helsinki language services. In addition, the text has been thoroughly edited by a native English speaker, Dr. Joel Levenson who is also an expert in therapeutic use of BCL-2 inhibitors.

NEW REFERENCES

IMpassion130: a Phase III randomized trial of atezolizumab with nab-paclitaxel for first-line treatment of patients with metastatic triple-negative breast cancer (mTNBC).

Leisha A. Emens, Sylvia Adams, Sherene Loi, Andreas Schneeweiss, Hope S. Rugo, Eric P. Winer, Carlos H. Barrios, Veronique Dieras, Juan de la Haba-Rodriguez, Luca Gianni, Stephen Y. Chui, and Peter Schmid
Journal of Clinical Oncology 2016 34:15_suppl, TPS1104-TPS1104

Inhibition of fatty acid oxidation as a therapy for MYC-overexpressing triple-negative breast cancer.

Camarda R, Zhou AY, Kohnz RA, Balakrishnan S, Mahieu C, Anderton B, Eyob H, Kajimura S, Tward A, Krings G, Nomura DK, Goga A.
Nat Med. 2016 Apr;22(4):427-32. doi: 10.1038/nm.4055. Epub 2016 Mar 7.

Priming BCL-2 to kill: the combination therapy of tamoxifen and ABT-199 in ER+ breast cancer.

Deng J, Letai A.
Breast Cancer Res. 2013;15(5):317.

A BH3 Mimetic for Killing Cancer Cells.

Green DR.

Cell. 2016 Jun 16;165(7):1560. doi: 10.1016/j.cell.2016.05.080.

PMID:27315468

Dual Faces of IFN γ in Cancer Progression: A Role of PD-L1 Induction in the Determination of Pro- and Antitumor Immunity.

Mandai M, Hamanishi J, Abiko K, Matsumura N, Baba T, Konishi I.

Clin Cancer Res. 2016 May 15;22(10):2329-34. doi: 10.1158/1078-0432.CCR-16-0224. Epub 2016 Mar 25.

IFN- α directly promotes programmed cell death-1 transcription and limits the duration of T cell-mediated immunity.

Terawaki S, Chikuma S, Shibayama S, Hayashi T, Yoshida T, Okazaki T, Honjo T.

J Immunol. 2011 Mar 1;186(5):2772-9. doi: 10.4049/jimmunol.1003208. Epub 2011 Jan 24.

Molecular and cellular insights into T cell exhaustion.

Wherry EJ, Kurachi M.

Nat Rev Immunol. 2015 Aug;15(8):486-99. doi: 10.1038/nri3862. Review.

OTHER CHANGES

DISCUSSION

Sentences added:

Page 18: Notably, no therapeutic potential was observed in two in vivo breast cancer models with low MYC expression status.

Page 19: In 60-day follow-up the tumors eventually reappeared but, nevertheless, the therapeutic action of ABv+anti-PD-1 treatment was superior to other simultaneous tested treatments, including combination of chemotherapy with anti-PD-1.

Subtitle modified:

ABn treatment induces massive apoptosis and inhibits tumor growth in Myc-induced mammary tumors concomitantly with T lymphocyte infiltration and exhaustion

Fig. 6I and 7F.

During the manuscript revision period we further analyzed immunoprofiling data and also omitted few samples with low cell counts from the earlier data set. These further analyses did not change main results or the previous conclusions.

Modified text:

Materials and Methods: Countbright beads (Life technologies) were added to the sample tubes and the cell counts were normalized to 2000 beads counted during the sample acquisition performed with FACSVerse BD. The data were analysed with FlowJo version 10.3. Results are shown using the absolute number of cells, generally excluding cells belonging to parental populations with less than 90 cells.

Figure legend 7F: Tumor immunoprofiles after the adjuvant treatment. The heatmap shows fold-changes compared to vehicle + IgG. N = 4 mice / group, except N = 1 in the biopsy. *These populations contained less than 90 cells.

OLD FIGURE 6I.

NEW FIGURE

OLD FIGURE 7F.

NEW FIGURE 7F.

REVIEWERS' COMMENTS:

Reviewer #1 (Remarks to the Author):

I am quite impressed with the new data comparing additional treatment combinations and prolonged therapy (i.e. extending the outcome to 60 days). I was also glad that the reviewers tested 2 additional models (both transgenic and a MYC-low transgenic model). I believe these additional experiments have substantially strengthened the claims of the manuscript. I am now fully in favor of publication.

Sincerely,

Andrei Goga, MD, PhD
Professor, UCSF

Reviewer #2 (Remarks to the Author):

This is an interesting report of broad interest. The authors have done an excellent job in responding to the critique.